# C9orf72-ALS human iPSC microglia are pro-inflammatory and toxic to co-cultured motor neurons via MMP9

Björn F. Vahsen [1,2], Sumedha Nalluru[1], Georgia R. Morgan[1], Lucy Farrimond[1,2], Emily Carroll[1,2], Yinyan Xu[1,2,3], Kaitlyn M. L. Cramb[2,4], Benazir Amein [1], Jakub Scaber [1,2], Antigoni Katsikoudi[2,5], Ana Candalija[1], Mireia Carcolé [6], Ruxandra Dafinca [1,2], Adrian M. Isaacs [6], Richard Wade-Martins [2,4], Elizabeth Gray[1], Martin R. Turner [1], Sally A. Cowley [7] ✉ & Kevin Talbot [1,2] ✉

Amyotrophic lateral sclerosis (ALS) is a neurodegenerative disease characterized by progressive motor neuron loss, with additional pathophysiological involvement of non-neuronal cells such as microglia. The commonest ALS-associated genetic variant is a hexanucleotide repeat expansion (HRE) mutation in *C9orf72*. Here, we study its consequences for microglial function using human iPSC-derived microglia. By RNA-sequencing, we identify enrichment of pathways associated with immune cell activation and cyto-/chemokines in C9orf72 HRE mutant microglia versus healthy controls, most prominently after LPS priming. Specifically, LPS-primed C9orf72 HRE mutant microglia show consistently increased expression and release of matrix metalloproteinase-9 (MMP9). LPS-primed C9orf72 HRE mutant microglia are toxic to co-cultured healthy motor neurons, which is ameliorated by concomitant application of an MMP9 inhibitor. Finally, we identify release of dipeptidyl peptidase-4 (DPP4) as a marker for MMP9-dependent microglial dysregulation in co-culture. These results demonstrate cellular dysfunction of C9orf72 HRE mutant microglia, and a non-cell-autonomous role in driving C9orf72-ALS pathophysiology in motor neurons through MMP9 signaling.

Amyotrophic lateral sclerosis (ALS) is primarily characterized by the degeneration of motor neurons (MNs) in the cortex, brainstem, and spinal cord. MN demise leads to progressive paralysis and premature death, typically within 3 years of symptom onset. ALS is thought to arise as a combination of cell-autonomous neuronal dysfunction and non-cell-autonomous factors driven by non-neuronal cells[1–3]. Microglia, the resident macrophages of the CNS parenchyma, are strongly associated with the complex neuroinflammatory process in ALS[4]. Microglia support neuronal homeostasis, for instance, by secreting cytokines and chemokines, pruning synapses, and phagocytosing

[1]Oxford Motor Neuron Disease Centre, Nuffield Department of Clinical Neurosciences, University of Oxford, John Radcliffe Hospital, Oxford OX3 9DU, UK. [2]Kavli Institute for Nanoscience Discovery, University of Oxford, Dorothy Crowfoot Hodgkin Building, Oxford OX1 3QU, UK. [3]Chinese Academy of Medical Sciences (CAMS), CAMS Oxford Institute (COI), Nuffield Department of Medicine, University of Oxford, Oxford OX3 7FZ, UK. [4]Oxford Parkinson's Disease Centre, Department of Physiology, Anatomy and Genetics, University of Oxford, Dorothy Crowfoot Hodgkin Building, Oxford OX1 3QX, UK. [5]Molecular Neurodegeneration Research Group, Nuffield Department of Clinical Neurosciences, University of Oxford, Dorothy Crowfoot Hodgkin Building, Oxford OX1 3QU, UK. [6]UK Dementia Research Institute at UCL and Department of Neurodegenerative Disease, UCL Queen Square Institute of Neurology, London WC1N 3BG, UK. [7]James and Lillian Martin Centre for Stem Cell Research, Sir William Dunn School of Pathology, University of Oxford, Oxford OX1 3RE, UK. ✉ e-mail: sally.cowley@path.ox.ac.uk; kevin.talbot@ndcn.ox.ac.uk

dead cells[5]. However, various stimuli including lipopolysaccharides (LPS) can induce microglial activation and a pro-inflammatory phenotype that can promote neurotoxicity[6]. In the context of ALS, widespread microglial activation has been particularly observed in patients with hexanucleotide repeat expansions (HRE) in *C9orf72*, the commonest genetic cause of ALS[7], and correlated with disease progression[8,9]. Mechanistically, in neurons, the HRE in *C9orf72* results in C9orf72 haploinsufficiency, the formation of RNA foci and dipeptide repeats (DPRs), and mislocalization of TDP-43[10,11]. C9orf72 is highly expressed in myeloid cells[12], but the consequences of the HRE in *C9orf72* for microglial function are less clear. C9orf72 loss-of-function through *C9orf72* knock-out (KO) resulted in a pro-inflammatory state in myeloid cells and microglia in mice[12,13]. In murine co-cultures, *C9orf72* KO microglia were toxic to neurons[14]. However, whether human *C9orf72* HRE mutant microglia display pathological features associated with the HRE, show a disease phenotype, or non-cell-autonomously affect neurons, is currently unclear.

We have previously established and extensively validated protocols for the differentiation of human induced pluripotent stem cell (iPSC)-derived microglia in monoculture and in co-culture with MNs, which closely resemble bona fide human microglia[15–17]. Here, we provide a comprehensive phenotypic analysis of human C9orf72 mutant iPSC microglia. RNA sequencing reveals a characteristic profile in C9orf72 mutant microglia, most prominently after priming with LPS, with enrichment of pathways associated with immune cell activation and chemo-/cytokines. Specifically, we uncover notably consistent upregulation of matrix metalloproteinase-9 (MMP9) in C9orf72 mutant microglia. In co-culture, unstimulated C9orf72 mutant microglia have a dysregulated cyto-/chemokine profile but do not significantly affect the activity or survival of healthy spinal MNs. However, after priming with LPS, C9orf72 mutant microglia are toxic to healthy spinal MNs, which is ameliorated by concomitant application of an MMP9 inhibitor. In addition, we identify MMP9-dependent release of dipeptidyl peptidase-4 (DPP4) as a marker for microglial dysfunction in co-culture.

## Results

### Differentiation of C9orf72-ALS patient-derived iPSC microglia

For the differentiation of iPSC microglia, we used our well-established EB-based protocol that generates microglia precursors of assumed yolk sac origin[15–17], resulting in close resemblance to bona fide microglial cells[16,17]. In this study, we generated iPSC microglia from three different C9orf72-ALS patients, three sex- and age-matched healthy donors, and one isogenic control line, previously obtained through CRISPR-mediated homology directed repair[18] (Fig. 1a). All lines readily differentiated into iPSC microglia, showing microglial morphology and high expression of the key microglial markers IBA1 and TMEM119 by immunofluorescent staining, Western blot, and RT-qPCR (Fig. 1b and Supp. Fig. 1a–f). Interestingly, the baseline expression of the homeostatic *P2RY12* gene was reduced in C9orf72 mutant microglia compared with controls (Supp. Fig. 1e). In all, 48 h of LPS treatment substantially decreased *P2RY12* expression, as expected (Supp. Fig. 1e). iPSC microglia generated from all lines were phagocytically competent using pHrodo zymosan particles, taken up by phagocytosis and degraded via lysosomes, as confirmed by treatment with Cytochalasin D and Bafilomycin A1, respectively (Supp. Fig. 1g, h). LPS priming increased the phagocytic uptake of pHrodo particles (Supp. Fig. 1h). Using an automated assessment of cell morphology[19], we found no morphological differences between C9orf72 mutant and healthy control microglia, but reduced cell volume and branch length compared with the isogenic control after LPS priming (Supp. Fig. 1i). The analysis of cell viability using an MTS assay showed no differences between C9orf72 mutant and control microglia (Supp. Fig. 2a). Together, these data demonstrate the successful differentiation of iPSC microglia from C9orf72-ALS and control lines.

### C9orf72-ALS iPSC microglia show pathological features associated with the HRE in *C9orf72*

We then analyzed C9orf72 mutant microglia for the presence of pathological features associated with the HRE in *C9orf72*, namely C9orf72 haploinsufficiency, expression of DPRs, RNA foci formation, and TDP-43 mislocalization. Western blotting demonstrated significantly higher C9orf72 expression in healthy microglia than in MNs (Fig. 1c), pointing towards a crucial role in microglial function. RT-qPCR for *C9orf72* and its variants 1 and 2/3 showed a significantly increased expression in LPS-primed microglia (Fig. 1d and Supp. Fig. 2b, c), which we confirmed by Western blot (Fig. 1e), implicating C9orf72 in the microglial response to pro-inflammatory stimulation. Compared with healthy microglia, however, C9orf72 mutant microglia showed significantly reduced C9orf72 expression, demonstrating C9orf72 haploinsufficiency in this comparison (Fig. 1e). The isogenic-C9-2 comparison showed no difference in C9orf72 levels (Fig. 1e).

Next, we examined C9orf72 mutant microglia for the presence of gain-of-function products of the HRE in *C9orf72*. The DPRs Poly(GA) and Poly(GP) were readily detectable by MSD ELISA[20] in C9orf72 mutant motor neurons and absent in healthy and isogenic controls (Fig. 1f, g). C9orf72 mutant microglia also expressed Poly(GA) and Poly(GP), at notably lower levels than MNs generated from the same lines (Fig. 1f, g). In all, 48 h and 6 d of LPS priming led to little change in Poly(GA) expression (Supp. Fig. 2d). Furthermore, we detected both sense and anti-sense RNA foci in C9orf72 mutant microglia using RNAscope, but not in healthy and isogenic controls (Fig. 1h). Interestingly, anti-sense RNA foci were present in significantly more microglial cells than sense foci. The number of sense and anti-sense foci per foci-positive nucleus did not show a statistically significant difference (Supp. Fig. 2e). Finally, quantification of the cytoplasmic/nuclear distribution of TDP-43 showed no differences between C9orf72 mutant microglia and controls (Supp. Fig. 2f), indicating the absence of TDP-43 mislocalization in C9orf72 mutant microglia.

Together, these data outline an important role for C9orf72 in microglia, and the presence of C9orf72 loss-of-function and HRE-associated gain-of-function products in mutant lines.

### C9orf72-ALS iPSC microglia have a dysregulated transcriptome with enrichment of pathways associated with immune cell activation, cytokines/chemokines, lysosomes, and the ER

Next, we performed RNA sequencing to investigate whether C9orf72 mutant iPSC microglia showed a mutation-associated transcriptional signature. We analyzed all six C9orf72-ALS and healthy control lines, unstimulated and LPS-primed, using three technical replicates per condition and line (Fig. 2a). Confirming our RT-qPCR data, we found clear expression of the microglial markers *TMEM119* and *P2RY12* as well as *TREM2*, *MERTK*, *C1QA*, *PROS1*, *GAS6*, and *GPR34*, with LPS leading to a notable reduction in expression of multiple genes (Supp. Fig. 3a). Principal component (PC) analysis showed a shift on PC2 between LPS-primed and unstimulated microglia (Fig. 2b and Supp. Fig. 3b). Several immune cell activation-associated pathways were enriched in LPS-primed versus unstimulated microglia by GSEA (Supp. Fig. 3c), confirming a clear response to LPS treatment.

On PC6, both stimulated and unstimulated C9orf72 mutant microglia separated out from healthy controls (Fig. 2b), indicating transcriptomic differences between the genotypes. We then performed differential expression analysis between C9orf72 mutant and control microglia for both unstimulated and LPS-stimulated comparisons, merging the three different differentiations for each line. We identified 34 differentially expressed genes (DEGs) for the unstimulated healthy-C9orf72 comparison, while LPS treatment increased the number of DEGs to 41 (Fig. 2c). 9 DEGs overlapped between the unstimulated and LPS-primed comparisons (Fig. 2c and Supp. Fig. 3d–g). Studying pathway enrichment, one immune

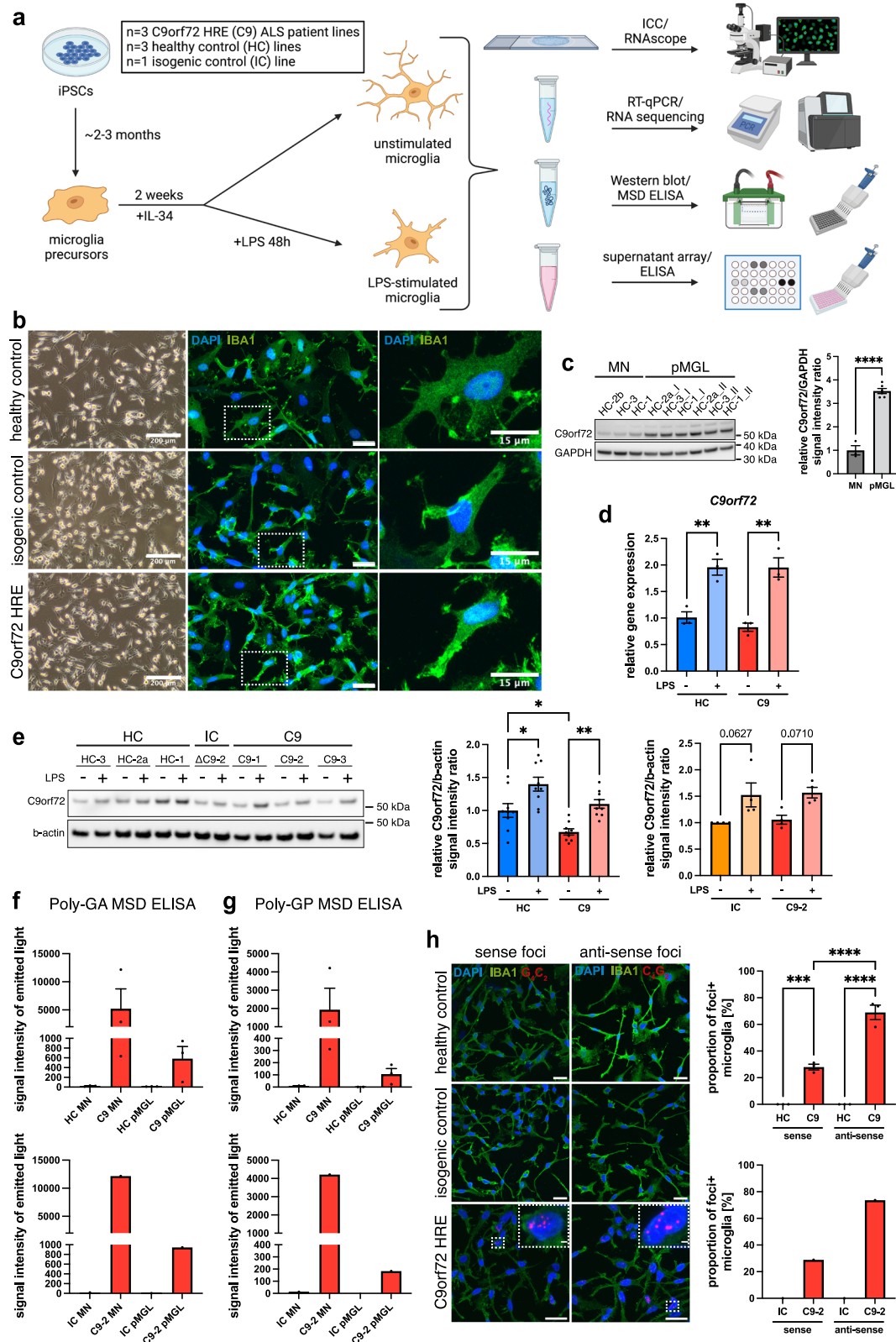

response term, but primarily development and chromatid-associated terms, were enriched in unstimulated C9orf72 mutant microglia by GSEA (Supp. Fig. 4a, b). However, after LPS treatment, over-representation analysis of the DEGs and GSEA identified enrichment of immune cell activation and cytokine/chemokine-associated pathways in LPS-stimulated C9orf72 mutant microglia relative to LPS-stimulated healthy controls, with a significant increase of the

chemokines *CXCL1* and *CXCL6* (Fig. 2d–f and Supp. Figs. 3g and 4c). In addition, we found negative enrichment for endoplasmic reticulum (ER)-associated terms in both unstimulated and LPS-primed C9orf72 mutant microglia by GSEA (Fig. 2f and Supp. Fig. 4b), while LPS-primed C9orf72 mutant microglia also negatively enriched lysosomes and several membrane-associated terms (Fig. 2f and Supp. Fig. 4d). Finally, we evaluated the DEGs between C9orf72 mutant and

**Fig. 1 | C9orf72-ALS patient-derived iPSC microglia show key pathological features associated with the hexanucleotide repeat expansion in *C9orf72*.**
**a** Schematic overview showing experimental setup and analyses performed on C9orf72 mutant (C9) iPSC-derived microglia in monoculture (pMGL) in this study, unstimulated or treated with LPS (100 ng/mL, 48 h). **b** Live cell (left) and immunofluorescent images (right, insets) of pMGL differentiated from healthy control (HC), isogenic control (IC), and C9 iPSC lines showing microglial morphology (images from one representative differentiation). Scale bars: 200 μm (left), 25 μm (right), 15 μm (insets). **c** Left: Western blot against C9orf72 comparing HC motor neurons (MNs) and pMGL. Right: Quantification normalized to the housekeeping gene GAPDH ($n = 3$ lines per cell type; $n = 1$ differentiation for MN, $n = 2$ differentiations for pMGL, indicated by I/II). **d** RT-qPCR for *C9orf72* in unstimulated and LPS-primed pMGL normalized to the housekeeping gene *TBP* ($n = 3$ lines per condition, $n = 1$ differentiation). **e** Left: exemplar Western blot against C9orf72 in unstimulated and LPS-primed HC, IC, and C9 pMGL. Right: quantification

normalized to the housekeeping gene b-actin. C9orf72 haploinsufficiency is found in C9 pMGL compared with HC ($n = 3$ lines per condition, $n = 3$ differentiations each) but not in the IC-C9-2 comparison ($n = 1$ line per condition, $n = 3$ differentiations each). **f, g** Analysis of Poly(GA)/Poly(GP) expression by MSD ELISA in HC, IC, and C9 MNs and pMGL. Both DPRs are detectable in C9 MNs and pMGL but not controls, while pMGL appear to have lower expression ($n = 1$ differentiation for $n = 3$ lines for top graphs, $n = 1$ line for bottom graphs). **h** Left: detection of nuclear sense ($G_4C_2$) and anti-sense ($G_2C_4$) RNA foci by RNAscope in C9 pMGL but not controls ($n = 1$ differentiation for $n = 3$ lines for top graph, $n = 1$ line for bottom graph). Scale bars: 25 μm, 2 μm (insets). Right: quantification showing anti-sense foci are more common than sense foci in C9 pMGL. Single data points and means ± SEM. *$P < 0.05$; **$P < 0.01$; ***$P < 0.001$; ****$P < 0.0001$. Two-tailed unpaired *t*-test (**c**). One-way ANOVA with Tukey's post-hoc test (**d, e, h**). No statistical tests for **f** and **g** due to the small sample size with high variability. The graphics for panel **a** were created using Biorender.com. Source data are provided as a Source Data file.

healthy control microglia using the different differentiations per lines as different datapoints. As expected, this analysis identified additional DEGs (Supp. Fig. 4e, f), notably, increased expression of the ALS-associated gene *CALHM2*[21] (Calcium Homeostasis Modulator Family Member 2) in C9orf72 mutant microglia.

In summary, these data reveal the dysregulation of several pathways, notably immune cell activation, cytokines/chemokines, lysosomes, and the ER, in C9orf72 mutant microglia (Fig. 2g).

## C9orf72-ALS iPSC microglia have a pro-inflammatory profile with consistently increased expression and release of MMP9

The strong enrichment of immune cell activation and cytokine/chemokine-associated pathways in our RNA-sequencing prompted us to evaluate the release of cyto-/chemokines from C9orf72 mutant microglia. Harnessing our isogenic-C9-2 line pair as a screening tool, we used a supernatant array to find dysregulated cyto-/chemokines in the culture supernatants. We identified differential expression of MMP9, GDF-15, and CXCL10 after LPS priming in our C9-2-isogenic pair (Supp. Fig. 5a). Whereas CXCL10 and GDF-15 were not consistently different in C9orf72 mutant microglia compared with isogenic and healthy controls (Supp. Fig. 5a–d), MMP9 was upregulated in both comparisons (Supp. Fig. 5a, b). MMP9 has previously been connected with neurodegeneration in murine SOD1 and TDP-43 ALS models[22,23], indicating that changes in its expression in C9orf72 mutant microglia are likely to be of biological relevance in ALS. We therefore focused on exploring MMP9 and confirmed significantly increased expression and activity of MMP9 in culture supernatants from LPS-primed C9orf72 mutant microglia versus both healthy and isogenic controls by ELISA and activity assay (Fig. 3a, b). Of note, MMP9 was not detectable by ELISA in culture supernatants from motor neurons generated from the same C9orf72 mutant iPSC lines (Supp. Fig. 5e). RT-qPCR and Western blot confirmed significantly increased pro- and active MMP9 expression at mRNA and protein level in C9orf72 mutant microglia compared with healthy controls (Fig. 3c, d). Similarly, the C9orf72-isogenic comparison showed increased expression at gene and protein level, reaching statistical significance for active MMP9 (Fig. 3c, d). Albeit less pronounced, MMP9 expression and release was also increased in C9orf72 mutant microglia after alternative priming with tumor necrosis factor (TNF) and Interleukin-1B (IL1B) (Supp. Fig. 5f, g). These data demonstrate that MMP9 expression in C9orf72 mutant microglia is consistently upregulated at mRNA and protein level resulting in increased secretion. Several other proteases and inhibitors in addition to MMP9 were increased in the culture supernatants from LPS-primed C9orf72 mutant microglia, including Cathepsin L, MMP-2, and TIMP-2 (Supp. Fig. 5h, j) using an array for various proteases and inhibitors. However, unlike MMP9, neither of these showed a corresponding upregulation in the RNA-sequencing (Supp. Fig. 5h–k). Taken together, these data identify consistently increased expression and release of MMP9 in C9orf72 mutant microglia.

## Unstimulated C9orf72-ALS iPSC microglia are not toxic to healthy motor neurons in co-culture

Having demonstrated a cell-autonomous microglial phenotype due to the HRE in *C9orf72*, we hypothesized that C9orf72 mutant microglia might exert a deleterious effect on spinal MNs. To concentrate on the non-cell-autonomous role of microglia in co-culture, we focused our investigations on co-cultures with healthy spinal MNs. Both cell types were co-cultured for at least 14 days (Fig. 4a), as described previously[16,17], speculating that overt differences would become apparent within this time frame. We seeded both cell types at a 1:1 ratio, mimicking the human glia:neuron ratio determined by a recent meta-analysis[24]. Immunocytochemical analysis confirmed a ~1:1 ratio of IBA1-positive microglia and TUJ1-positive neurons in co-culture irrespective of the microglial genotype (Fig. 4b and Supp. Fig. 6a/b). The neuronal population in co-culture contained ~80% ChAT-positive spinal MNs (Supp. Fig. 6c). In agreement, microglial IBA1 and MN ChAT protein levels were consistently high by Western blot in co-culture, with IBA1 absent in MN monoculture (Supp. Fig. 6d, e). Morphological analysis showed no differences between C9orf72 mutant microglia and controls in co-culture (Supp. Fig. 6f).

We then proceeded to assess MMP9 expression in co-culture. In keeping with our monoculture experiments, pro-MMP9 was significantly upregulated in co-cultures with C9orf72 mutant microglia compared with healthy controls by Western blotting, and a similar trend was observed in comparison with isogenic microglia (Supp. Fig. 6d, g). Active MMP9 was not differentially expressed, likely due to the lack of LPS stimulation (Supp. Fig. 6d, g). MN monocultures showed no expression of MMP9, confirming its microglial origin (Supp. Fig. 6d). Expression of the apoptotic marker cleaved caspases 3 (CC3) was low, and equal between genotypes in co-culture, and neuron numbers were similar (Fig. 4c and Supp. Fig. 6b), indicating that unstimulated C9orf72 mutant microglia did not cause overt neurotoxicity. Co-cultures showed the expected neuronal properties using patch-clamping electrophysiology, but no significant differences in the neuronal resting membrane potential, capacitance, sodium and potassium currents, or action potential properties (Fig. 4d, e and Supp. Fig. 7a, b). Similarly, we observed no differences in the neuronal mean firing rate using MEA recordings nor regarding neurite outgrowth, NfL release, and synaptophysin expression (Fig. 4f–h and Supp. Fig. 7c, d). In summary, these data demonstrate that unstimulated C9orf72 mutant microglia do not alter the survival or function of two-week-co-cultured MNs.

## Co-cultures of unstimulated C9orf72-ALS iPSC microglia with MNs show an altered supernatant profile and upregulate DPP4

We next proceeded to assess the supernatant profile of the unstimulated co-cultures. We hypothesized that a dysregulated profile

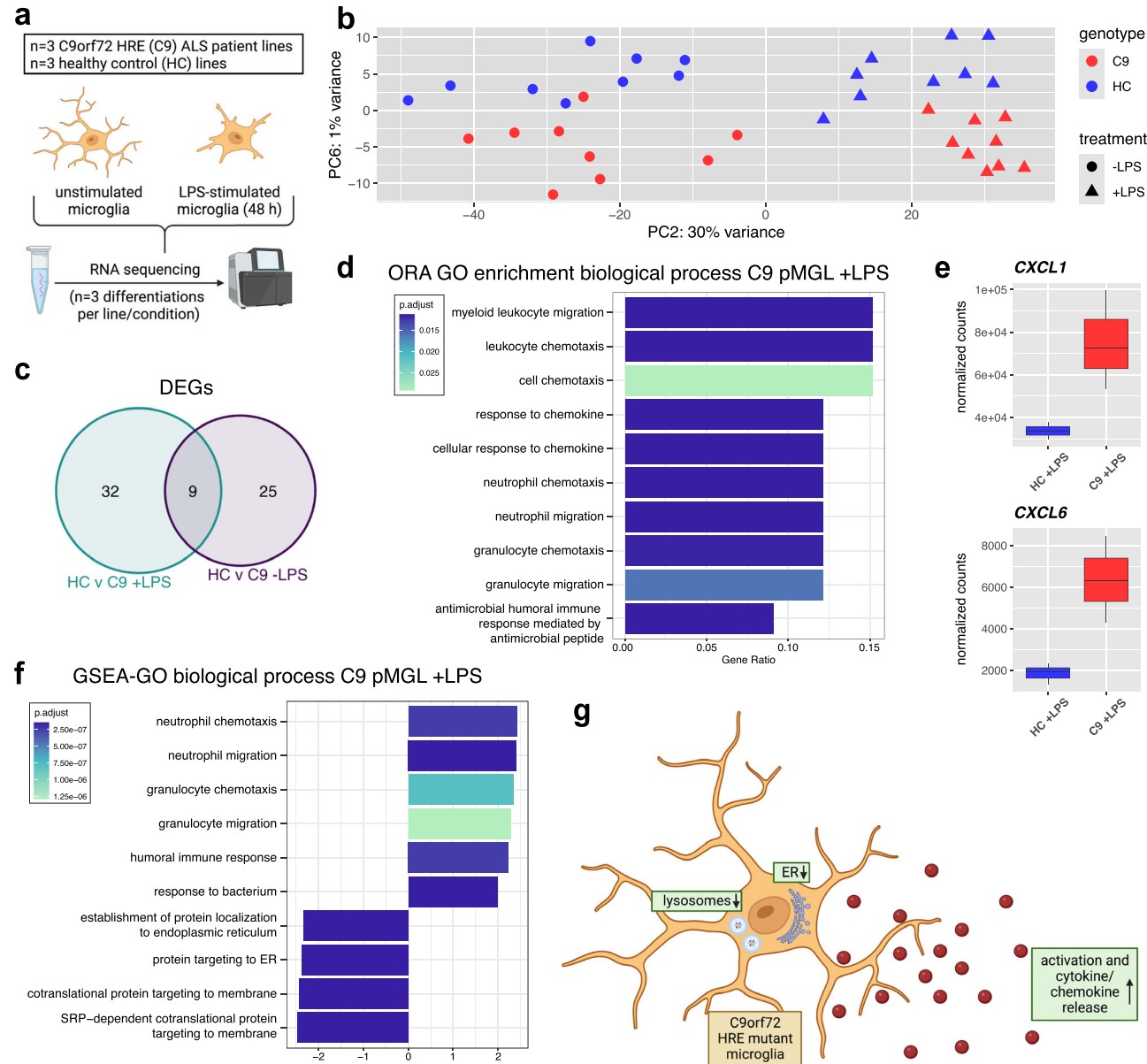

**Fig. 2 | RNA sequencing of C9orf72 mutant iPSC microglia identifies dysregulation of pathways associated with immune cell activation, cyto-/-chemokines, lysosomes, and the ER. a** Schematic overview showing the experimental setup for RNA sequencing of C9orf72 mutant (C9) and healthy control (HC) iPSC-derived microglia in monoculture (pMGL), unstimulated or treated with LPS (100 ng/mL, 48 h). **b** Principal component analysis (PCA) plot based on the top 500 most variable genes with biggest variance in unstimulated and LPS-stimulated (100 ng/mL, 48 h) HC and C9 pMGL showing PC2 and PC6 ($n = 3$ lines per condition, $n = 3$ differentiations each), with response to LPS treatment on PC2 and separation of genotypes on PC6. For downstream analyses shown in this figure, the different differentiations for each line were merged. **c** Venn diagram showing differentially expressed genes (DEGs), defined as |log₂ fc| > 0.5 and adjusted *p*-value < 0.05, and overlap for unstimulated and LPS-stimulated comparisons. **d** Bar plots showing top 10 significantly enriched GO biological process terms after over-representation

analysis (ORA) using the clusterProfiler package for the DEGs between LPS-primed C9 and HC pMGL. x-axis showing Gene Ratio. The Benjamini-Hochberg corrected p-values for pathway enrichment for each term is indicated by the color of the bars. **e** Box plots of normalized counts from RNA seq analysis for *CXCL1* and *CXCL6*, two DEGs in LPS-primed C9 pMGL versus HC ($n = 3$ lines per condition). Box plots show the median (center line), upper and lower quartiles (box limits), 1.5x interquartile range (whiskers), and outliers (points). **f** Bar plots showing top 10 significantly different GO biological process terms for LPS-stimulated C9 pMGL versus HC based on gene set enrichment analysis (GSEA) using the clusterProfiler package with the whole transcriptome ranked by log₂fc. x-axis showing normalized enrichment score (NES). The Benjamini−Hochberg corrected p-values for pathway enrichment for each term is indicated by the color of the bars. **g** Summary schematic illustrating the pathways with most prominent enrichment in C9 pMGL in the RNA seq analysis. The graphics for panel **a**, **g** were created using Biorender.com.

might serve as an early marker for microglial dysregulation which could translate into overt effects on MNs upon prolonged culture duration or stimulation. Indeed, several neurotrophic factors and pro-inflammatory molecules were dysregulated, with the strongest increase for dipeptidyl peptidase 4 (DPP4), comparing co-cultures of C9orf72 mutant microglia with healthy and isogenic controls (Fig. 4i). Notably, CXCL1, one of the DEGs increased in C9orf72 mutant

microglia in our RNA-sequencing (Fig. 2e), showed the second highest increase comparing C9orf72 mutant and healthy control microglia (Fig. 4i). We focused on exploring DPP4, which has previously been implicated in neurodegeneration in Alzheimer's disease[25] and, intriguingly, DPP4 shedding has been connected with MMP9[26]. ELISA validation confirmed significant upregulation of DPP4 in co-cultures with C9orf72 mutant microglia compared with both

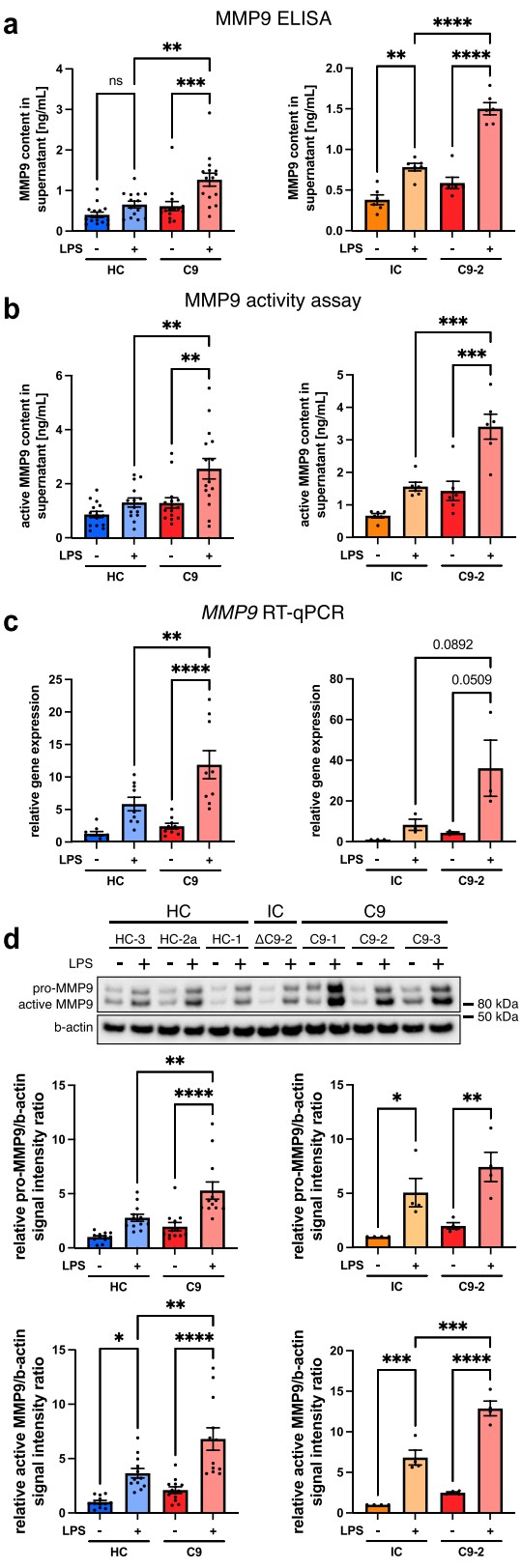

**Fig. 3 | C9orf72 mutant iPSC microglia are pro-inflammatory with consistently upregulated expression and release of MMP9. a** ELISA quantification of MMP9 in supernatants from unstimulated and LPS-stimulated healthy control (HC), isogenic control (IC), and C9orf72 mutant (C9) microglia in monoculture (pMGL) shows significantly increased MMP9 release in the C9-HC (*n* = 3 lines per condition, *n* = 5 differentiations each) and IC-C9-2 pMGL comparisons (*n* = 1 line per condition, *n* = 6 differentiations each). **b** Fluorometic activity assay for active MMP9 in supernatants from unstimulated and LPS-stimulated HC, IC, and C9 pMGL shows significantly increased MMP9 activity in the C9-HC (*n* = 3 lines per condition, *n* = 5 differentiations each) and IC-C9-2 pMGL comparison (*n* = 1 line per condition, *n* = 6 differentiations each). **c** RT-qPCR for *MMP9* expression in unstimulated and LPS-stimulated HC, IC, and C9 pMGL normalized to the housekeeping gene *TBP* showing significantly increased *MMP9* expression in the C9-HC (*n* = 3 lines per condition, *n* = 3 differentiations each) and numeric increase in IC-C9-2 pMGL comparison (*n* = 1 line per condition, *n* = 3 differentiations each). **d** Top: exemplar Western blot against the pro-form and active form of MMP9 in unstimulated and LPS-stimulated HC, IC, and C9 pMGL. Bottom: quantification shows significantly increased expression of pro-MMP9 in the C9-HC pMGL comparison and a numeric increase in the IC-C9-2 pMGL comparison. Active MMP9 is significantly increased in both comparisons. Both normalized to the housekeeping gene b-actin (*n* = 3 lines per condition for C9-HC comparison, *n* = 1 line for the IC-C9-2 comparison, *n* = 4 differentiations each). Single data points and means ± SEM. *\*P* < 0.05; *\*\*P* < 0.01; *\*\*\*P* < 0.001; *\*\*\*\*P* < 0.0001. One-way ANOVA with Tukey's post-hoc test (**a–d**). Source data are provided as a Source Data file.

demonstrate an altered cytokine profile of unstimulated C9orf72 mutant microglia in co-culture with MNs, in absence of overt neurotoxicity, and identify DPP4 release as a putative biomarker for early microglial dysfunction.

### LPS-primed C9orf72-ALS iPSC microglia increase apoptotic marker expression in co-cultured healthy MNs via an MMP9-dependent mechanism

The dysregulated cytokine profile in unstimulated co-cultures and the phenotype of LPS-stimulated microglial monocultures led us to speculate that pro-inflammatory priming might uncover non-cell-autonomous effects of C9orf72 mutant microglia on spinal MNs. Microglia are exquisitely sensitive to LPS, with high levels of CD14 and TLR4 and well-described cytokine responses[16]. Therefore, to investigate this hypothesis, we stimulated co-cultures repetitively with LPS (3 doses over 10 days) (Fig. 5a). The resulting stimulated co-cultures showed similar microglia:neuron ratios between genotypes, with ~30% microglia (Fig. 5b and Supp. Fig. 8a, b). The MN marker ChAT and the microglia marker IBA1 were expressed in co-culture by Western blotting, with IBA1 expression absent in MN monocultures (Fig. 5c and Supp. Fig. 8c, d). The neuronal population in co-culture contained ~80% ChAT-positive spinal MNs (Supp. Fig. 8e). Both pro- and active MMP9 were significantly upregulated in stimulated co-cultures with C9orf72 mutant microglia compared with healthy and isogenic microglia (Fig. 5c, d), in line with our experiments on LPS-primed microglial monocultures (Fig. 3). MN monocultures showed no expression of MMP9, confirming its microglial origin (Fig. 5c).

Upon LPS treatment, MNs in co-culture with stimulated C9orf72 mutant microglia showed significantly higher neuronal expression of CC3 compared with healthy controls by immunostaining (Fig. 5e, f), indicating that LPS-primed C9orf72 mutant microglia induced apoptosis in MNs in co-culture. This effect was phenocopied by the C9-2-isogenic comparison but did not reach statistical significance (Fig. 5e, f). MNs co-cultured with C9orf72 mutant microglia in inserts showed significantly reduced viability by MTS assay, suggesting a soluble mechanism of microglial neurotoxicity (Supp. Fig. 8f). The number of MNs was equal in all conditions (Supp. Fig. 8b), suggesting that the activation of neuronal apoptotic pathways did not yet lead to substantial loss of neurons at this time point. Neuronal activity, the expression of synaptophysin, NfL release, and neurite outgrowth did not show significant differences (Supp. Fig. 9a–d).

healthy and isogenic controls (Fig. 4j). In contrast, supernatants from microglial monocultures showed no significant differences between C9orf72 mutant microglia and controls (Supp. Fig. 7e). Supernatants from MNs derived from the same C9orf72 iPSC lines contained almost no DPP4, and DPP4 showed a punctate staining pattern localizing to microglia in co-culture (Supp. Fig. 7f, g), suggesting that DPP4 is released from microglia in co-culture. Together, these data

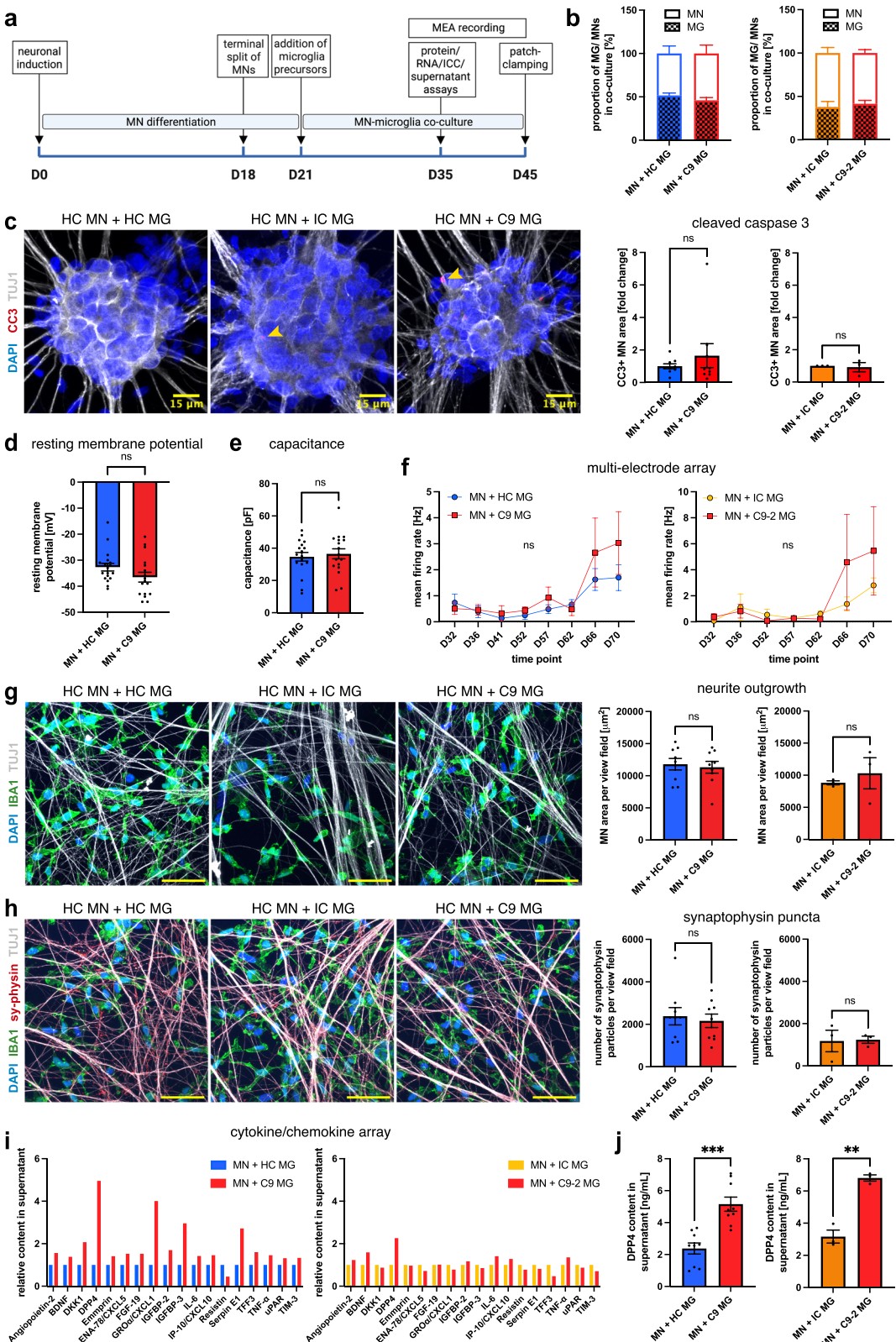

We then assessed whether increased microglial expression of MMP9 contributed to these neurotoxic effects. As proof of principle, administration of recombinant human MMP9 reduced MN viability measured by MTS assay, which was rescued by concomitant treatment with an MMP9 inhibitor that has previously been successfully applied in vivo[22] (Supp. Fig. 9e). Indeed, simultaneous administration of LPS and the MMP9 inhibitor reduced CC3 expression in co-cultured MNs,

abolishing the significant difference between co-cultures with C9orf72 mutant and healthy microglia (Fig. 5e, f). The same pattern was evident in the C9-2-isogenic comparison, albeit not reaching statistical significance (Fig. 5e, f). Treatment of co-cultured control microglia with the MMP9 inhibitor resulted in no differences compared with DMSO-treated controls, supporting a specific effect of MMP9 inhibition on C9orf72 mutant microglia.

**Fig. 4 | Unstimulated C9orf72 mutant iPSC microglia do not cause overt toxicity to co-cultured healthy motor neurons but show an altered cytokine profile and upregulate DPP4 release. a** Experimental setup for motor neuron (MN)-microglia (MG) co-culture. **b** Percentage of MNs/MG in co-cultures of healthy control (HC) MNs and HC, isogenic control (IC), and C9orf72 mutant (C9) MG showing *a* - 1:1 ratio (*n* = 3 microglial lines (C9-HC comparison), *n* = 1 microglial line (IC-C9-2 comparison), *n* = 3 differentiations). **c** Left: Exemplar images showing the apoptosis marker cleaved caspase 3 (CC3) in HC MNs in co-culture with MG. Scale bars: 15 μm. Right: Quantification of relative fold change of CC3 expression (*n* = 3 microglia lines (C9-HC comparison), *n* = 1 microglial line (IC-C9-2 comparison), *n* = 3 differentiations). **d, e** Patch clamping analysis of resting membrane potential and capacitance of HC MNs in co-culture with MG (datapoints are single MNs from co-cultures with *n* = 3 microglial lines (**d**, +HC: *n* = 18, +C9: *n* = 17; **e**, both *n* = 17), *n* = 1 differentiation. **f** Multi-electrode array analysis (MEA) of mean firing rate of HC MNs in co-culture with MG (*n* = 3 microglial lines (C9-HC comparison), *n* = 1 microglial line (IC-C9-2 comparison), *n* = 2 wells per *n* = 3 differentiations). **g** Left:

exemplar images of neurites of HC MNs in co-culture with MG. Right: quantification of neurite outgrowth (*n* = 3 microglial lines (C9-HC comparison), *n* = 1 microglial line (IC-C9-2 comparison), *n* = 3 differentiations). Scale bars: 50 μm. **h** Left: exemplar images of synaptophysin (sy-physin) staining in co-culture of HC MNs with MG. Right: quantification of the number of synaptophysin particles (*n* = 3 microglial lines (C9-HC comparison), *n* = 1 microglial line (IC-C9-2 comparison), *n* = 3 differentiations). Scale bars: 50 μm. **i** Relative fold change in co-culture supernatant for selected targets using a cytokine array (pooled from co-cultures of HC MNs with *n* = 3 microglial lines (C9-HC comparison), *n* = 1 microglial line (IC-C9-2 comparison), *n* = 3 differentiations). **j** ELISA quantification of DPP4 in co-culture supernatants from HC MNs and MG (*n* = 3 microglial lines for C9-HC comparison, *n* = 1 microglial line for IC-C9-2 comparison, *n* = 3 differentiations). Single data points and means ± SEM. **P < 0.01; ***P < 0.001; ns not significant. Two-tailed unpaired *t*-test (**c–e, g, h, j**), two-way ANOVA with Tukey's post-hoc test (**f**). The graphics for panel a were created using Biorender.com. Source data are provided as a Source Data file.

---

These data suggest that the increased expression of MMP9 in C9orf72 mutant microglia causes non-cell-autonomous toxicity to motor neurons in co-culture.

### DPP4 release into the culture supernatant is an MMP9-dependent marker of microglial dysfunction in co-culture

We then evaluated the supernatant cytokine profile in co-cultures with LPS-stimulated C9orf72 mutant microglia and found an upregulation of IL6, MCP-3, MIP-1α/β, MIP-3α, and DPP4, amongst others, compared with healthy controls (Fig. 5g). Intriguingly, concomitant administration of the MMP9 inhibitor decreased the release of several cyto-/chemokines and, importantly, reduced the expression of DPP4 and Endoglin only in co-cultures with C9orf72 but not control microglia (Fig. 5g), suggesting a specific effect on C9orf72 mutant microglia. We confirmed significantly increased release of DPP4 in co-cultures with C9orf72 mutant microglia compared with healthy and isogenic controls by ELISA, with MMP9 inhibition significantly reducing DPP4 release (Fig. 5h). DPP4 release into the culture supernatant therefore directly reflects MMP9 activity, with DPP4 release as an MMP9-dependent marker of microglial dysfunction in co-culture.

### C9orf72-ALS iPSC microglia cause neurodegeneration of co-cultured healthy MNs after long-term LPS exposure via an MMP9-dependent mechanism

Finally, we assessed if the C9orf72 mutant microglia-mediated activation of neuronal apoptotic pathways might cause substantial loss of neurons after prolonged pro-inflammatory treatment. We exposed co-cultures to a more frequent and longer LPS treatment paradigm (10 doses over 20 days) in the absence of BDNF and GDNF, which might counteract neuronal cell death (Fig. 6a). Again, C9orf72 mutant microglia increased the neuronal expression of CC3 compared with healthy controls, which was ameliorated by concomitant administration of the MMP9 inhibitor (Fig. 6b). Furthermore, C9orf72 mutant microglia caused a significant reduction in the number of spinal MNs after prolonged LPS exposure compared with healthy control microglia (Fig. 6c and Supp. Fig. 9f). Administration of the MMP9 inhibitor ameliorated neuron loss, abolishing the significant difference between co-cultures with C9orf72 mutant and healthy microglia (Fig. 6c). These data demonstrate that C9orf72 mutant microglia cause overt toxicity to spinal MNs after chronic pro-inflammatory stimulation with LPS.

## Discussion

Here, we demonstrate the successful differentiation of human iPSC microglia from C9orf72-ALS patients. Using RNA sequencing, we show enrichment of pathways associated with immune cell activation, cyto-/chemokine activity, lysosomes, and the ER, most prominently after priming with LPS. Specifically, C9orf72 mutant microglia have a pro-inflammatory phenotype with consistent upregulation and release of

MMP9. In iPSC-microglia-iPSC-MN co-culture experiments, unstimulated C9orf72 mutant microglia have a dysregulated supernatant profile but do not affect the activity or survival of healthy spinal MNs. However, after LPS priming, C9orf72 mutant microglia induce apoptotic signaling in healthy MNs in co-culture and cause overt neurodegeneration after chronic LPS exposure. Mechanistically, concomitant application of an MMP9 inhibitor ameliorates the neurotoxic properties of LPS-stimulated C9orf72 mutant microglia. Finally, we identify elevated DPP4 release from C9orf72 mutant microglia into the co-culture supernatant as a marker of microglial dysregulation, which decreases in response to MMP9 inhibitor treatment.

In this study, we aimed to answer several key questions about the cell-autonomous and non-cell-autonomous consequences of the *C9orf72* HRE in microglia. Previous studies[12–14] have demonstrated an important role for C9orf72 in macrophages in mice, with *C9orf72* KO inducing a pro-inflammatory profile. However, *C9orf72* KO does not recapitulate gain-of-function effects of the *C9orf72* HRE, and whether authentic human iPSC microglia from C9orf72-ALS patients replicate these findings remains an open question. To address this, we analyzed human iPSC microglia from three different C9orf72-ALS patients compared with three different healthy individuals, and additionally used our recently generated isogenic line as another control[18]. Of note, two independent studies studying C9orf72 mutant iPSC microglia have recently been published[27,28]. There is consensus where our analyses overlap, which we discuss alongside our data below.

First, we demonstrated C9orf72 haploinsufficiency versus healthy controls, expression of the DPRs Poly(GA) and Poly(GP), and formation of sense and anti-sense RNA foci in C9orf72 mutant microglia. These data show that both loss-of-function and gain-of-function toxicity of the *C9orf72* HRE is present in C9orf72 mutant microglia in agreement with Lorenzini et al.[27] and Banerjee et al.[28]. Interestingly, DPRs and RNA foci were detectable in C9orf72 mutant microglia but not in healthy and isogenic controls, whereas only the C9orf72-healthy comparison but not the C9orf72-isogenic pair demonstrated C9orf72 haploinsufficiency. Similarly, we had found no differences in C9orf72 expression at protein level for MNs generated from this isogenic line pair in our previous study[18]. While this observation could be related to epigenetic changes triggered by the mutation that are not reverted upon expansion removal, it allowed us to assess in our phenotypic analyses whether mutation-induced microglial phenotypes were observable in the presence of gain-of-function toxicity only. Notably, even though C9orf72 expression was significantly higher in human iPSC-microglia than in iPSC-MNs, thereby confirming the finding from O'Rourke et al.[12] in mice, we showed that Poly(GA)/(GP) expression appears to be higher in C9OF72 MNs than in C9orf72 mutant microglia. This observation suggests that microglia might have an intrinsic capacity, through a mechanism currently unknown, for dealing with DPR accumulation, a phenomenon that should be explored in future

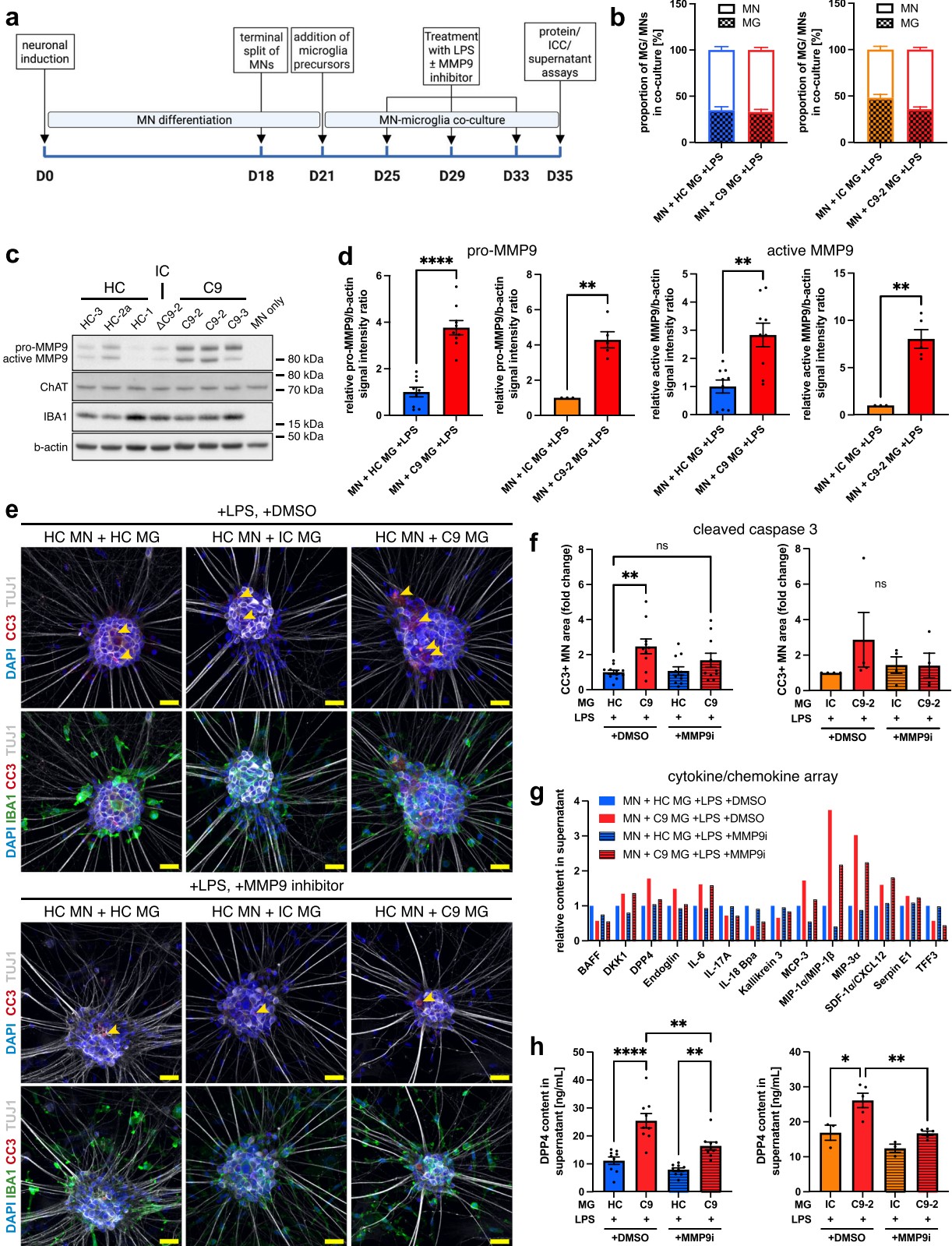

studies. However, we did not observe TDP-43 mislocalization in C9orf72 mutant microglia, corroborating the observations by Lorenzini et al.[27].

Interestingly, we found that C9orf72 expression increased in microglia treated with LPS, implicating C9orf72 in the microglial response to pro-inflammatory stimulation. By RNA sequencing, we uncovered a dysregulated transcriptome in C9orf72 mutant microglia,

with the biologically most relevant changes after stimulation with LPS, including differential expression of the chemokines *CXCL1* and *CXCL6*. Specifically, we showed positive enrichment of pathways associated with immune cell activation and cytokines/chemokines in C9orf72 mutant microglia, while terms associated with lysosomes and the ER, amongst others, were negatively enriched. In comparison, Lorenzini et al.[27] reported few transcriptomic differences between

**Fig. 5 | LPS-stimulated C9orf72 mutant iPSC microglia increase apoptotic marker expression in co-cultured healthy iPSC motor neurons and upregulate DPP4 release via an MMP9-dependent mechanism. a** Experimental setup for LPS-stimulated motor neuron (MN)-microglia (MG) co-cultures. **b** Percentage of MNs/MG in LPS-stimulated co-cultures of healthy control (HC) MNs and HC, isogenic control (IC), and C9orf72 mutant (C9) MG showing *a* - 30% microglia content (*n* = 3 microglial lines, *n* = 4 differentiations (C9-HC comparison), *n* = 1 microglial line, *n* = 5 differentiations (IC-C9-2 comparison)). **c** Exemplar Western blot against IBA1, ChAT, and MMP9 in LPS-stimulated co-cultures of HC MNs with MG. **d** Quantification of blot shown in **c** demonstrating pro-MMP9 and active MMP9 normalized to the housekeeping gene b-actin are significantly increased in LPS-stimulated co-cultures of HC MNs with C9 MG compared with controls (*n* = 3 microglial lines (C9-HC comparison), *n* = 1 microglial line (C9-2-IC comparison), *n* = 3 differentiations). **e** Exemplar images of apoptosis marker cleaved caspase 3 (CC3) in HC MNs in LPS-primed co-cultures with MG plus MMP9 inhibitor 1 (3 μM, MMP9i) or DMSO as vehicle control. Scale bars: 25 μm. **f** Quantification showing significantly increased relative fold change in CC3 expression in LPS-primed co-cultures with C9 MG compared with HC MG. Administration of MMP9i ameliorates microglial neurotoxicity. The C9-2-IC comparison shows the same pattern but does not reach statistical significance (*n* = 3 microglial lines, *n* = 4 differentiations (C9-HC comparison), *n* = 1 microglial line (IC-C9-2 comparison)). **g** Relative fold change in LPS-stimulated co-culture supernatant for selected targets using a cytokine array. Supernatant samples from LPS-primed co-cultures of HC MNs with C9 MG showing a dysregulated cytokine profile compared with HC MG, which is ameliorated by MMP9i treatment (pooled samples from *n* = 3 microglial lines with *n* = 3 differentiations each). **h** ELISA quantification of DPP4 in co-culture supernatants from LPS-stimulated co-cultures of HC MNs and MG showing significantly increased DPP4 levels, which is significantly reduced after MMP9i treatment (*n* = 3 microglial lines (C9-HC comparison), *n* = 1 microglial line (IC-C9-2 comparison), *n* = 3 differentiations). Single data points and means ± SEM. *P < 0.05; **P < 0.01; ****P < 0.0001; ns: not significant. Two-tailed unpaired *t*-test (**d**) and one-way ANOVA with Tukey's post-hoc test (**f**, **h**). The graphics for panel a were created using Biorender.com. Source data are provided as a Source Data file.

LPS-stimulated C9orf72 mutant microglia and controls, possibly due to the shorter LPS treatment (6 h vs 48 h in this study). C9orf72 has been previously shown to localize to endosomes, lysosomes, and phagosomes in macrophages[29]. Loss-of-function of C9orf72, on the other hand, has been connected with disturbance of the endo-lysosomal and autophagy pathways in neurons[30,31], murine macrophages/microglia[12,14], and, recently, iPSC microglia[28] and could be the underlying reason for the negative enrichment in lysosome-associated pathways in C9orf72 mutant microglia. Similarly, previous evidence connects the *C9orf72* HRE with ER stress in neurons[32], warranting further exploration of the negative enrichment of ER-associated terms in C9orf72 mutant microglia.

In this study, we focused on exploring the association with C9orf72 mutant microglia and cytokines/chemokines. Most strikingly, we identified consistently increased MMP9 expression and release from LPS-primed C9orf72 mutant microglia. MMP9 is an endopeptidase that can cleave several constituents of the ECM and cell surface receptors[33,34]. MMP9 dysregulation is common to several neurodegenerative disorders[34], and a neurotoxic role for MMP9 has been demonstrated using SOD1 and TDP-43 animal models for ALS[22,23]. Intriguingly, MMP9 was independently identified as one of the interactors of C9orf72 in control iPSC microglia[28]. However, to our knowledge, MMP9 upregulation in C9orf72 mutant microglia has not been described previously. Interestingly, MMP9 was significantly increased in C9orf72 mutant microglia compared with both healthy and isogenic controls, and thus also in the absence of C9orf72 haploinsufficiency in the C9-2-isogenic comparison. It is therefore likely that MMP9 upregulation is not a consequence of reduced C9orf72 levels but rather of gain-of-function products of the *C9orf72* HRE such as DPRs or RNA foci. This is of particular interest considering a pro-inflammatory microglial phenotype has thus far been primarily linked with C9orf72 loss-of-function based on results from *C9orf72* KO in murine microglia/macrophages[12,14] and, recently, iPSC microglia[28].

We then hypothesized that C9orf72 mutant microglia might cause direct neurotoxicity. We limited our experimental design to healthy MNs, to exclude confounding effects of the simultaneous presence of the HRE in *C9orf72* in MNs and to focus on primary neurotoxic sequelae of *C9orf72* HRE in microglia. In unstimulated co-cultures, neuron survival and activity remained unchanged. However, upon repetitive priming of co-cultures with LPS, C9orf72 mutant microglia induced apoptotic markers in MNs and, after prolonged exposure, overt neurodegeneration, demonstrating direct neurotoxic effects of C9orf72 mutant microglia. Our findings suggest that C9orf72 mutant microglia possess latent neurotoxicity, requiring a pro-inflammatory environment to promote their damaging properties. It is possible that C9orf72 astrocytes, which show multiple facets of dysfunction[4], or other immune cells provide these pro-inflammatory factors. Banerjee et al.[28] similarly demonstrated neurotoxicity driven by C9orf72 mutant microglia in co-cultures with MNs after an excitotoxic treatment with AMPA, supporting the notion that stimulation is required to uncover the latent neurotoxic properties of C9orf72 mutant microglia. In contrast with previous experiments showing that *C9orf72* KO microglia promote synapse loss[14], we found no effect of C9orf72 human iPSC microglia on multiple readouts of synaptic protein expression or MN activity, even after LPS stimulation. A limitation of our study is the comparably short co-culture interval, and it is possible that synaptic deficits might become apparent after prolonged co-culture, which might also uncover non-cell-autonomous microglial toxicity without the requirement for external pro-inflammatory stimuli.

Most importantly, we confirmed that MMP9 is consistently upregulated in C9orf72 mutant microglia-MN co-cultures and showed that the neurotoxic effect of C9orf72 mutant microglia is directly mediated by MMP9, as treatment with an MMP9 inhibitor ameliorated apoptotic marker expression and neurodegeneration in co-cultured MNs. In mice, MMP9 expression has previously been found to be substantially higher in ALS-vulnerable (spinal and fast) MNs compared with relatively resistant (oculomotor and slow) MNs[22]. Inhibition of MMP9 was beneficial in a mutant SOD1[G93A] model for ALS, using the same MMP9 inhibitor we used here[22], and a TDP-43 based mouse model for sporadic ALS[23], suggesting MMP9 as a therapeutic target. However, MMP9 inhibition in these studies was limited to MNs or generically affected all cells in CNS. Here, we therefore provide evidence for MMP9 inhibition as a therapeutic strategy specifically focusing on microglia and C9orf72-ALS. A neuroprotective effect could either be mediated through direct inhibition of soluble MMP9 or through an indirect influence on MMP9-dependent cellular phenotypes in C9orf72 mutant microglia.

Finally, we also investigated the release of potential biomarkers from C9orf72 mutant microglia. One key candidate which requires further investigation is MMP9 itself, previously shown to be increased in the serum of ALS patients[35] and in the spinal cord and skin of mutant SOD1[G93A] transgenic mice[36]. In addition, we identified DPP4 release from C9orf72 mutant microglia as an early event in co-culture with MNs. DPP4 is a peptidase that has previously been implicated in neurodegeneration in Alzheimer's disease[37], with DPP4 inhibition having shown beneficial outcomes in mouse models[25], but it has not previously been connected with ALS. Importantly, DPP4 levels were also upregulated in supernatants from unstimulated co-cultures – in absence of overt microglial toxicity – and decreased in response to MMP9 inhibitor treatment in LPS-simulated cultures, indicating its potential to serve as an early marker for microglial dysfunction. DPP4 shedding by MMP9 has been demonstrated in adipocytes[26] and could represent the underlying mechanism. Hence, MMP9 and DPP4 levels should be analyzed in the biofluids of C9orf72-ALS patients to

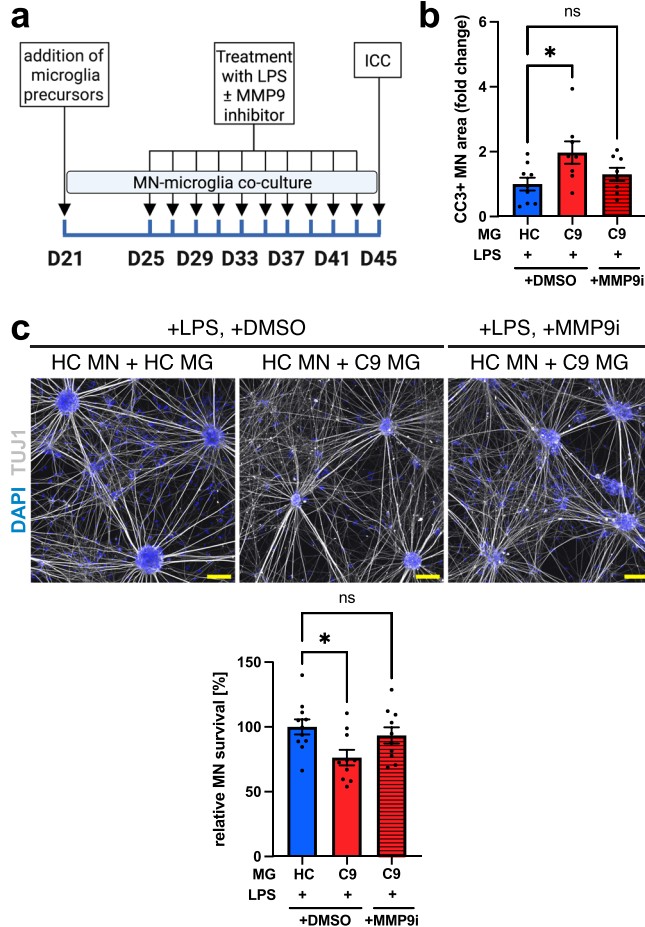

**a**

addition of microglia precursors

Treatment with LPS ± MMP9 inhibitor

ICC

MN-microglia co-culture

D21   D25 D29 D33 D37 D41 D45

**b**

CC3+ MN area (fold change)

| MG | HC | C9 | C9 |
|----|----|----|----|
| LPS | + | + | + |
| | +DMSO | | +MMP9i |

ns / *

**c**

+LPS, +DMSO

HC MN + HC MG    HC MN + C9 MG

+LPS, +MMP9i

HC MN + C9 MG

DAPI TUJ1

relative MN survival [%]

ns / *

| MG | HC | C9 | C9 |
|----|----|----|----|
| LPS | + | + | + |
| | +DMSO | | +MMP9i |

**Fig. 6 | C9orf72 mutant iPSC microglia cause non-cell-autonomous toxicity to co-cultured healthy iPSC motor neurons after prolonged LPS treatment via an MMP9-dependent mechanism. a** Experimental setup for experimental setup for prolonged treatment of motor neuron (MN)-microglia (MG) co-cultures with LPS. **b** Quantification showing significantly increased relative fold change in cleaved caspase 3 (CC3) expression in co-cultures of healthy control (HC) MNs with C9orf72 mutant (C9) MG compared with HC MG after prolonged LPS treatment. Treatment with MMP9 inhibitor 1 (3 μM, MMP9i) ameliorates microglial neurotoxicity (n = 3 microglial lines per genotype, n = 3 differentiations). **c** Top: exemplar images showing co-cultures of HC MNs with HC and C9 MG after pro-longed LPS treatment. Bottom: quantification showing significantly reduced relative MN survival in co-cultures of HC MNs with C9 MG compared with HC MG after prolonged LPS treatment. Treatment with MMP9i ameliorates microglial neurotoxicity (n = 3 microglial lines per genotype, n = 4 differentiations). Scale bars: 100 μm. Single data points and means ± SEM. *$P < 0.05$; ns: not significant. One-way ANOVA with Tukey's post-hoc test (**b, c**). The graphics for panel a were created using Biorender.com. Source data are provided as a Source Data file.

explore their potential as biomarkers in disease stratification and treatment trials.

A question that currently remains unanswered is the response of C9orf72 mutant microglia to mutant C9orf72-associated primary neuronal dysfunction in MNs, which was beyond the scope of this study. The findings by Banerjee et al.[28] suggest that C9orf72 mutant microglia are equally toxic to C9orf72 mutant MNs, at least after excitotoxic insult with AMPA. It will be particularly interesting to examine the effect of microglia on aggregated TDP-43 or DPRs in C9orf72 mutant neurons, to see if microglia can provide neuroprotection against protein aggregates in co-culture, as suggested by an animal model for TDP-43 proteinopathy[38], or whether neuronal aggregates will activate C9orf72 mutant microglia and potentiate their non-cell-autonomous toxicity.

In summary, we provide evidence for dysfunction of C9orf72 mutant microglia, and primary neurotoxicity of C9orf72 mutant microglia mediated by MMP9 as a therapeutic target to counteract neurodegeneration in C9orf72-ALS. We identify MMP9-dependent DPP4 release as a putative biomarker for microglial dysfunction in culture, paving the way for future studies that assess the levels of MMP9 and DPP4 in the biofluids of C9orf72-ALS patients.

## Methods

### Ethics statement
All iPSC lines were previously derived from skin biopsy fibroblasts, following signed informed consent, collected under ethical approval granted by the South Wales Research Ethics Committee (WA/12/0186) and the South Central Berkshire Research Ethics Committee (REC10/H0505/71) in the James Martin Stem Cell Facility, University of Oxford, under standardized protocols. The iPSC lines have all been published previously (see below and Supplementary Table 1).

### iPSC culture
Three iPSC lines derived from three different ALS patients carrying a HRE in *C9orf72* and as controls, an isogenic line, previously generated through CRISPR/Cas9 genome editing using homology directed repair[18], and four sex- and age-matched healthy control iPSC lines, were used in this study. All iPSC lines have previously been published and extensively characterized (for details on demographics and characterization, see Supplementary Table 1). iPSC were cultured in mTeSR™1 (85850, StemCell Technologies) on Geltrex™ (A1413302, ThermoFisher) with daily medium changes. Cells were passaged using EDTA (0.5 mM), expanded to produce consistent, frozen cell stocks for the study (cryopreserved in 50% mTeSR™1, 30% embryonic stem cell fetal bovine serum (16141002, Merck), 10% KnockOut™ DMEM (10829018, ThermoFisher), 10% DMSO (D2650, Merck)) with a maximum of 3 subsequent passages during experimentation, and tested negative for mycoplasma using the MycoAlert™ Mycoplasma Detection Kit (Lonza).

### Differentiation of iPSC-derived motor neurons
iPSC were differentiated to MNs using a previously published protocol[39]. Briefly, neural induction was performed using DMEM-F12/Neurobasal 50:50 supplemented with N2 (1X), B27 (1X), 2-Mercaptoethanol (1X), Antibiotic-Antimycotic (1X, all ThermoFisher), Ascorbic Acid (0.5 μM), Compound C (1 μM, both Merck), and Chir99021 (3 μM, R&D Systems). After two days in culture, Retinoic Acid (1 μM, Merck) and Smoothened Agonist (500 nM, R&D Systems) were added. After another 2 days, Compound C and Chir99021 were removed. On day in vitro (DIV)18, the cells were re-plated on polyethylenimine (PEI, 0.07%, Merck) and Geltrex™ (ThermoFisher) or PDL (Sigma-Aldrich)/ Laminin (R&D Systems)/ Fibronectin (Corning) in medium additionally supplemented with BDNF (10 ng/mL), GDNF (10 ng/mL), Laminin (0.5 μg/mL, all ThermoFisher), and DAPT (10 μM, R&D Systems). Three days later, the medium was changed to microglia medium as described below, and MNs were either co-cultured with microglia or maintained in monoculture. Half medium changes were performed every 2–4 days.

### Differentiation of iPSC-derived microglia precursors
iPSC were differentiated to macrophage/microglia precursors as described previously[15,40]. Briefly, embryoid body (EB) formation was induced by seeding iPSC into Aggrewell 800 wells (STEMCELL Technologies) in mTeSR™1 supplemented with BMP4 (50 ng/mL), VEGF (50 ng/mL, both Peprotech), and SCF (20 ng/mL, Miltenyi Biotec). After four to seven days with daily ¾ medium changes, EBs were transferred to T175 flasks and differentiated in X-VIVO15 (Lonza), supplemented with Interleukin-3 (25 ng/mL, R&D Systems), MCS-F (100 ng/mL), GlutaMAX (1X, both ThermoFisher), and 2-Mercaptoethanol (1X). Fresh medium was added weekly. After approximately two months,

precursors were harvested by collecting the supernatant, passed through a cell strainer (40 μM, Falcon/Greiner), and either differentiated to microglia in monoculture or co-culture as described below.

## Differentiation of iPSC-derived microglia

Microglia precursors were plated on PEI (0.07%) and Geltrex™ or PDL/Laminin/Fibronectin using our previously characterized microglia medium[17] comprised of Advanced DMEM-F12 (ThermoFisher), Gluta-MAX (1X), N2 (1X), Antibiotic-Antimycotic (1X), 2-Mercaptoethanol (1X), Interleukin-34 (100 ng/mL, Peprotech/BioLegend), BDNF (10 ng/mL), GDNF (10 ng/mL), and Laminin (0.5 μg/mL). Microglia were differentiated for approximately 14 days, with half medium changes every 2–4 days, and collected for RNA/protein extraction or used for assays.

## Co-culture of iPSC-derived motor neurons and microglia

Co-cultures were generated as described before using the HC-2b line (Supplementary Table 1) to generate MNs[17]. Briefly, three days after the final re-plating of differentiating MNs (DIV21), microglia precursors were harvested. MNs were rinsed with PBS, and microglia precursors re-suspended in microglia medium were added to each well in a 1:1 ratio with the MNs. Co-cultures were maintained for at least 14 days, with half medium changes every 2–4 days. In select experiments, direct contact of the two cell types was prevented by plating microglia in tissue culture inserts (0.4 μm pore size, 83.3932.040, Sarstedt).

## Priming of microglia with LPS and MMP9 inhibitor treatment

For microglial monocultures, all medium was aspirated, and LPS (100 ng/mL, L4391, Merck) in microglia medium was added for 48 h. Control wells received fresh microglia medium only. In select conditions, repetitive LPS treatments were performed on day 0, 2, and 4. For co-cultures, LPS treatments were started on DIV25 and performed for 10 days. Half of the medium was aspirated and replaced with LPS in microglia medium (final concentration: 100 ng/mL), with additional half medium changes with LPS-containing microglia medium on DIV29 and 33. In additional experiments, LPS treatments were started on DIV25 and performed for 20 days, with half media changes every other day. BDNF and GDNF were removed from the culture media for these experiments. To inhibit MMP9, cultures were concomitantly treated with LPS (100 ng/mL) and MMP9 inhibitor I (MMP9i, 3 μM, 444278, Merck) or DMSO in equal concentrations (0.3%) as vehicle control.

## Priming of microglia with TNF/IL1B

All medium was aspirated from microglial monocultures, and TNF (50 ng/mL) and IL1B (20 ng/mL, both Peprotech) in microglia medium were added for 48 h. Control wells received fresh microglia medium only.

## Treatment of MNs with rhMMP9 and MMP9 inhibitor

MNs were treated with recombinant active human MMP9 (rhMMP9, 30 ng/mL, PF024-5UG, Merck) and MMP9i (3 μM, 444278, Merck) or DMSO in equal concentrations (0.3%) as vehicle control. Treatments were performed on DIV21 and DIV25, and MN viability was determined by MTS assay on DIV29.

## Live-imaging of phagocytosis and quantification

Unstimulated and LPS-primed (100 ng/mL for 48 h) microglial monocultures were stained with Hoechst in Live Cell Imaging Solution (1:10,000, ThermoFisher) for 10 min. pHrodo™ Red Zymosan Bioparticles™ (P35364, ThermoFisher) in Live Cell Imaging Solution (50 μg/mL) were added to each well and images were acquired after 150 min using an EVOS™ XL Core Cell Imaging System (ThermoFisher) equipped with a 10x objective. In selected conditions, cells were additionally treated with the phagocytosis/degradation inhibitors Cytochalasin D (10 μM, C2618, Merck), Bafilomycin A1 (1 μM, 1334, R&D Systems), or DMSO (vehicle), with a 1 h-pretreatment. Quantification

was performed automatically using an analysis macro in Fiji and the area of phagocytosis particles per cell was obtained by dividing the area of the internalized phagocytosis particles by the number of Hoechst-positive microglia.

## Patch-clamp electrophysiology

Whole-cell patch-clamp electrophysiology was performed as described previously[17]. Healthy control MNs on DIV 42–46 in co-culture with healthy control or C9orf72-ALS patient-derived microglia were maintained in an extracellular solution containing 167 mM NaCl, 2.4 mM KCl, 1 mM MgCl₂, 10 mM glucose, 10 mM HEPES, and 2 mM CaCl₂ adjusted to a pH of 7.4 and 300 mOsm. Electrodes with tip resistances between 7 and 12 MΩ were produced from borosilicate glass (0.86 mm inner diameter; 1.5 mm outer diameter). An intracellular solution was added to the electrode containing 140 mM K-Gluconate, 6 mM NaCl, 1 mM EGTA, 10 mM HEPES, 4 mM MgATP, 0.5 mM Na₃GTP, adjusted to pH 7.3 and 290 mOsm. Data acquisition was performed using a Multiclamp 700B amplifier, digidata 1550 A and clampEx 6 software. Series resistance ($R_s$) was maintained at <30 MΩ. Voltage gated channel currents were measured on voltage clamp where neurons were pre-pulsed for 250 ms with −140 mV pulse and a 10 mV-step voltage was applied from −70 mV to +70 mV. Induced action potentials (APs) were recorded on current clamp, neurons were held at −70 mV and 15 current steps of 10 pA were applied from −10 pA to 130 pA. Voltage gated channel currents and properties of induced action potentials (rheobase, max APs, APs over rheobase) were quantified using Clampfit 10.3 (pCLAMP Software suite, Molecular Devices). Max APs were quantified by determining the number of APs in sweep with the longest train. AP duration and amplitude were quantified in Matlab R2022a (MathWorks) using the first action potential recorded from each neuron.

## MEA recordings

In all, 2–15 min recordings of neuronal activity were obtained for co-cultures in 48-well multi-electrode array (MEA) plates (M768-tMEA-48B, Axion BioSystems) using the Maestro system (Axion Biosystems). Analysis was performed using the AXIS software v2.5.2 (Axion BioSystems) with SD > 5.5 as the detection threshold. The mean firing rate per well was calculated by averaging all active (>1 spike/min) electrodes per well.

## RNA isolation and RT-qPCR

RNA was extracted using an RNAeasy Mini Plus kit (Qiagen) and equal amounts of RNA (100 ng) were reverse-transcribed to cDNA using the High-Capacity cDNA Reverse Transcription Kit (ThermoFisher). Quantitative real-time PCR was performed with Fast SYBR™ Green Master Mix (ThermoFisher) using a LightCycler® 480 PCR System (Roche) and the primers (ThermoFisher) listed in Supplementary Table 3. All kits were used according to the manufacturer's instructions. Quantification of the relative fold gene expression was performed using the $2^{-\Delta\Delta Ct}$ method with normalization to the *TBP* reference gene (*TBP* Ct range 0.5) and the mean gene expression of the control microglia for each experiment.

## RNA sequencing

Three healthy control and three C9orf72-ALS patient-derived iPSC lines were differentiated into microglia, with three independent differentiations per line. RNA was extracted as described above. All RNA samples displayed RIN values of at least 8.5. Preparation of samples and bioinformatical analysis was then performed as described before[17]. Briefly, samples were normalized to 100 ng prior to library preparation. Polyadenylated transcript enrichment and strand specific library preparation was completed using the NEBNext Ultra II mRNA kit (NEB) following the manufacturer's instructions. Paired end sequencing (read length: 150 base pairs) was performed using a

NovaSeq6000 platform (Illumina, NovaSeq 6000 S2/S4 reagent kit, v1.5, 300 cycles), generating a raw read count of a minimum of 30 M reads per sample. Paired-end reads were mapped to the human GRCh38.p13 reference genome (https://www.gencodegenes.org) using HISAT2 v2.2.1[41]. The counts table was obtained using Feature-Counts v2.0.1[42]. Normalization of counts and differential expression analysis was performed using DESeq2 v1.28.1[43] in RStudio 1.4.1103, including the biological sex in the model and with the Benjamini-Hochberg method for multiple testing correction. The three different differentiations per condition were collapsed into one datapoint unless indicated otherwise where the biological sex and differentiation replicate were included in the model. Exploratory data analysis was performed following variance-stabilizing transformation of the counts table, using principal component analysis. Log$_2$ fold change (log$_2$ fc) shrinkage was performed using the 'normal' approach. Genes with $|\log_2 fc| > 0.5$ and adjusted $p$-value $< 0.05$ were defined as differentially expressed genes (DEG). Data was interpreted with annotations from the Gene Ontology database using clusterProfiler v3.16.1[44]. We analyzed pathway enrichment by performing over-representation analysis (ORA) for all DEGs and gene set enrichment analyses (GSEA) on the whole transcriptome ranked by log$_2$fc, using the standard settings in clusterProfiler for ORA and GSEA (adjusted $p$-value $< 0.05$ using the Benjamini-Hochberg method).

### Immunofluorescence
Cells cultured on coverslips were pre-fixed with 2% paraformaldehyde (PFA) in PBS for 2 min at room temperature (RT) and then fixed with 4% PFA in PBS for another 15 min at RT. After permeabilization and blocking with 5% donkey serum and 0.2% Triton X-100 in PBS for 1 h at RT, the coverslips were incubated with the primary antibodies listed in Supplementary Table 2, diluted in 1% donkey serum and 0.1% Triton X-100 in PBS, at 4 °C ON. After three washes with PBS-0.1% Triton X-100 for 5 min each, coverslips were incubated with corresponding fluorescent secondary antibodies Alexa Fluor® 488/568/647 donkey anti-mouse/rabbit/goat/guineapig (all 1:1000, ThermoFisher or Abcam). Coverslips were then washed twice with PBS-0.1% Triton X-100 for 5 min each and incubated with DAPI (1 μg/mL, Sigma-Aldrich) in PBS for 10 min. After an additional 5 min-washing step with PBS-0.1% Triton X-100, the coverslips were mounted onto microscopy slides using Pro-Long™ Diamond, Gold or Glass Antifade Mountant (ThermoFisher). Confocal microscopy was performed using an LSM 710, LSM 880 (both Zeiss) or Fluoview FV1000 (Olympus) microscope.

### Analysis of microglial ramifications
Four to five z-stack images (z-interval: 1 μm) of randomly selected visual fields were taken at 60x magnification for each coverslip, and maximum intensity projections were generated in Fiji. To analyze the branching of IBA1-positive microglia in monoculture and co-culture, the cell volume, ramification index, average branch length, and number of branch points was automatically determined using 3DMorph as described elsewhere[19].

### Quantification of TDP-43 mislocalization in microglia
Four to five z-stack images of randomly selected visual fields were taken at 60x magnification for each coverslip, and maximum intensity projections were generated in Fiji. The mean intensity values for TDP-43 in the nucleus and cytoplasm were determined in a blinded fashion, with the DAPI and IBA1 signal used to identify the nuclear and cytoplasmic boundaries, respectively. The ratio of cytoplasmic to nuclear TDP-43 was then calculated.

### Quantification of microglia marker expression in microglia monocultures
Three z-stack images of randomly selected visual fields were taken at 40x magnification for each coverslip, and maximum intensity

projections were generated in Fiji. The number of DAPI-positive nuclei and IBA1$^+$/DAPI$^+$ and TMEM119$^+$/DAPI$^+$ cells was automatically quantified per view field for each line using a macro.

### Quantification of microglia and motor neuron numbers and ratios in co-culture
Three to five z-stack images of randomly selected visual fields were taken at 10x or 40x magnification for each coverslip, and maximum intensity projections were generated in Fiji. For microglia, using a macro, an auto threshold was applied on the IBA1 channel to create a mask, and the number of microglial cells was determined for each view field by counting the number of DAPI-positive nuclei within the IBA1 mask using the 'analyze particles' function. For MNs, an auto threshold was applied on the TUJ1 channel to create a mask, and the number of neurons was determined for each view field by counting the number of DAPI-positive nuclei within the TUJ1 mask using the 'analyze particles' function. To calculate the microglia:MN ratio, the number of microglial cells was divided by the MN number. For the assessment of MN survival after long-term LPS exposure, the absolute MN number was normalized to the mean of the control lines to determine the relative MN survival by accounting for batch effects between differentiations after long-term culture.

### Quantification of apoptotic marker expression in co-culture
Five z-stacks images of randomly selected visual fields were taken at 40x magnification for each coverslip. Maximum intensity projections were generated. Using a macro, an auto threshold was applied on the TUJ1 channel to create a mask, and the expression of the CC3 apoptotic marker within neurons was determined for each view field by quantifying the area of the CC3 signal within the TUJ1 mask using the 'analyze particles' function. To account for batch effects between differentiations, the fold change in expression was generated by normalizing the absolute measurements to the mean of the control lines.

### Quantification of neurite outgrowth in co-culture
Four to five z-stacks of randomly selected view fields showing neurites but avoiding neuronal somas were obtained at 60x magnification for each coverslip. Maximum intensity projections were generated in Fiji. Using a macro, an auto threshold was applied to the TUJ1 staining, and the neurite outgrowth for each view field was determined by measuring the area of TUJ1-positive neurites per view field using the 'analyze particles' function in Fiji.

### Analysis of neuronal synaptophysin expression
Four to five z-stacks images of randomly selected visual fields were taken at 60x magnification for each coverslip. Maximum intensity projections were generated in Fiji. Using a macro, an auto threshold was applied to the synaptophysin staining to analyze dot-like structures, and the number and area of synaptophysin particles was determined for each view field using the 'analyze particles' function in Fiji.

### RNAscope
RNA foci were detected using the RNAscope Multiplex Fluorescent v2 Assay (323100, ACD) according to the manufacturer's instructions. In brief, iPSC microglia fixed with 4% PFA were sequentially dehydrated by 50%, 70% and 100% ethanol for 1 min each, and stored at −20 °C. After rehydrating with 100%, 70% and 50% ethanol, the cells were incubated with 0.1% Tween-20 in PBS, hydrogen peroxide and Protease III (dilution ratio 1:15) for 10 min each, with PBS washes in between. Probes targeting the (G$_4$C$_2$)$_n$ and (C$_4$G$_2$)$_n$ expansions (ACD, 884351-C2 and 884361-C2) were added to separate coverslips and incubated for 2 h, followed by amplification and development steps according to the standard protocol. Sequential ICC was performed by incubation with primary antibody against IBA1 (1:200, 019-19741, Wako Fujifilm) for 2 h

at RT and then according to the immunofluorescence protocol detailed above.

## Quantification of RNA foci formation in microglia

Five z-stack images of randomly selected visual fields were taken at 40x or 63x magnification for each coverslip, and maximum intensity projections were generated in Fiji. The number of foci-positive cells and number of foci per foci-positive nucleus were determined by manual counting.

## MTS viability assay

In select experiments, MN viability was determined using the Cell-Titer 96® AQueous One Solution Cell Proliferation Assay (MTS, G3580, Promega) according to the manufacturer's instructions. Plate readings were performed after 90–120 min incubation with assay reagents.

## Measurement of cytokine/chemokine and neurofilament release

Culture supernatants were collected and spun down at 1200 x g and 4 °C for 10 min. Pooled samples were analyzed using the Proteome Profiler Human XL Cytokine Array (ARY022B) or the Protease/Protease Inhibitor Array (ARY025, both R&D Systems) according to the manufacturer's instructions. The signal was visualized on a ChemiDoc™ MP imaging system (Bio-Rad) and analyzed using ImageStudioLite v5.2.5 (LI-COR). In addition, the Human DPP4 (DC260B), MMP9 (DMP900), CXCL10 Quantikine ELISAs (DIP100), and Active MMP9 Fluorokine E Kit (F9M00, all R&D Systems) were used according to the manufacturer's instructions. Human neurofilament light chain (NfL) release was measured using a Meso Scale Discovery (MSD) assay (K1517XR-2) according to the manufacturer's instructions.

## Protein extraction

Cells were rinsed with PBS, and 100–200 µL of cold RIPA buffer (ThermoFisher) supplemented with complete™ protease inhibitor cocktail and PhosSTOP™ phosphatase inhibitor (both Merck) were added to each well. 10 1s-impulses with motor-driven pestles were applied per sample to homogenize the lysates. After incubation on ice for 30 min, samples were centrifuged at 12,000 rpm and 4ºC for 15 min, and the supernatants were stored at −80 °C. Quantification of the protein concentration was performed using the Pierce™ Bicinchoninic Acid Protein Assay kit (ThermoFisher) according to the manufacturer's instructions. Protein samples were then prepared at defined concentrations (0.3–1.0 µg/µL) by dilution in distilled water, NuPAGE™ loading buffer and NuPAGE™ sample reducing agent (both ThermoFisher). This was followed by incubation at 95 °C for 5 min to denature and reduce the samples.

## Western blot

5–20 µg of each sample and Spectra Multicolour Broad Range Protein Ladder or PageRuler™ Prestained Protein Ladder (both ThermoFisher) were loaded onto pre-cast NuPAGE™ Bis-Tris 4–12% gradient gels (ThermoFisher). The samples were separated by sodium dodecyl sulfate polyacrylamide gel electrophoresis (SDS-PAGE) using NuPAGE™ MES running buffer (ThermoFisher) at 50–130 V. Proteins were then transferred to nitrocellulose membranes at 20–25 V for 7 min using an iBlot 2 Dry blotting system (both ThermoFisher). Following blocking with 5% milk or BSA in TBS/0.1% Tween-20 (TBS-T, ThermoFisher) at RT for one hour, membranes were incubated with the primary antibodies listed in Supplementary Table 4 diluted in 3% milk or BSA in TBS-T at 4 °C overnight. Membranes were washed three times with TBS-T for 7 min and then incubated with corresponding horseradish peroxidase (HRP)-coupled secondary antibodies at room temperature for 1 h. The following HRP-coupled secondary antibodies (all 1:5000) were used: sheep anti-mouse HRP (NA931V), donkey anti-rabbit HRP (NA934V, both GE Healthcare), donkey anti-goat HRP (PA1-28664,

ThermoFisher). Membranes were again washed three times with TBS-T for 7 min and then incubated for one minute with enhanced chemiluminescence solution (Millipore). The ChemiDoc™ MP imaging system was used to visualize the signal. Band intensities were quantified using Fiji v1.53q[45] and normalized to the housekeeping genes β-actin or GAPDH and the mean expression of the controls for each experiment. After development, membranes were washed with TBS-T two times for 5 min, followed by a 10 min-incubation with Restore™ PLUS stripping buffer (ThermoFisher). After two additional 5 min-washes with TBS-T, membranes were blocked and incubated with new primary antibodies as described above.

## Analysis of Poly(GA)/(GP) expression

Cells were lysed in RIPA buffer (Sigma-Aldrich) supplemented with 2× complete Mini, EDTA free protease inhibitor cocktail (Merck), and 2% SDS (ThermoFisher) and homogenized by sonication. After centrifugation at 17,000 × g at 16 °C for 20 min, the protein content of the supernatants was determined as described in the Supplementary Methods. The samples were adjusted to the same concentration (0.6–1.0 mg/mL) and MSD immunoassay was performed in singleplex using 96-well SECTOR plates (MSD) to quantify Poly(GA)/(GP) expression, as described previously[20]. In brief, plates were coated with unlabeled anti-poly(GP) or anti-poly(GA) antibodies. After blocking, samples were loaded at 45 µg of protein per well for the poly(GP) immunoassay and at 27 µg of protein for poly(GA). Biotinylated anti-poly(GP) and anti-poly(GA) antibodies were used as detectors, followed by sulfo-tagged streptavidin (Meso Scale Discovery, R32AD). Plates were read with the MSD reading buffer (Meso Scale Discovery, R92TC) using the MSD Sector Imager 2400. Signals correspond to intensity of emitted light upon electrochemical stimulation of the assay plate. Prior to analysis, the average reading from a calibrator containing no peptide was subtracted from each reading. Negative signals were set to '0'.

## Statistics and reproducibility

No statistical method was used to predetermine sample size. Investigators were blinded to allocation during analyses if permitted by the experimental design. Statistical analyses were conducted using GraphPad Prism 9. No data were excluded from the analyses. Data from multiple lines and differentiations were pooled to compare the overall effect of the *C9orf72* HRE versus controls. Comparisons of two groups were performed by two-tailed unpaired t-tests and multiple group comparisons by one-way or two-way ANOVA with appropriate post-hoc tests as indicated in the figure legends. The number of independent experiments and lines are indicated in each figure legend. Data are presented as single data points and means ± SEM. Differences were considered significant when $P < 0.05$ (*$P < 0.05$; **$P < 0.01$; ***$P < 0.001$; ****$P < 0.0001$; ns: not significant). Box plots show the median (center line), upper and lower quartiles (box limits), 1.5x interquartile range (whiskers), and outliers (points). GraphPad Prism 9 or RStudio 1.4.1103 were used to plot data. Final assembly of figures was done using Adobe Illustrator 25.4.1.

## Reporting summary

Further information on research design is available in the Nature Portfolio Reporting Summary linked to this article.

## Data availability

The RNA seq data generated in this study have been deposited in the GEO repository under accession code GSE217625. Paired-end reads were mapped to the human GRCh38.p13 reference genome (https://www.gencodegenes.org). All data supporting the findings of this study are available within the paper and its Supplementary Information. Source data are provided with this paper.

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

## Acknowledgements

We thank the Oxford Genomics Centre at the Wellcome Centre for Human Genetics (funded by Wellcome Trust grant reference 203141/Z/ 16/Z) for the generation and initial processing of the sequencing data. We thank Dr Elisa Giacomelli (Memorial Sloan Kettering Cancer Center, US) for helpful discussions about the data, Dr Hector Dejea (Lund University, Sweden) for help with the analysis of electrophysiology data, Dr Tina Wei and Dr Zameel Cader (University of Oxford, UK) for providing the MEA setup, and Dr Jennifer Davies (University of Oxford, UK) for help with NfL measurements. This work was supported by studentship awards from the University of Oxford Clarendon Fund and St John's College Oxford to B.F.V. and K.M.L.C., the Oxford-Medical Research Council (MRC) Doctoral Training Partnership (MR/N013468/1) to B.F.V. and E.C., the National Institute for Health Research (NIHR) Oxford Biomedical Research Centre to BFV, the Chinese Scholarship Council (CSC) and the Chinese Academy of Medical Sciences (CAMS) Innovation Fund for Medical Science (CIFMS), China (2018-I2M-2-002) to Y.X., the Natural Sciences and Engineering Research Council of Canada (PGSD3-517039-2018) to K.M.L.C., the Canadian Centennial Scholarship Fund to K.M.L.C., and a Wellcome Trust Doctoral Training Fellowship (102176/Z/ 13/Z) to L.F.; Further support came from the Motor Neurone Disease Association (MND Association) (project grant Talbot/Apr22/889-791) to B.F.V., K.T., and S.A.C., an MND Association Walker Professorship to M.R.T., the Academy of Medical Sciences (SGL025\1095) to J.S., the UK Dementia Research Institute, which receives its funding from UK DRI Ltd, funded by the UK MRC, Alzheimer's Society and Alzheimer's Research UK, to AMI, the Monument Trust Discovery Award (J-1403) from Parkinson's UK and the MRC Dementias Platform UK Stem Cell Network (Grant MR/M024962/1) to RWM, and the Oxford Martin School to S.A.C. The views expressed are those of the authors and not necessarily those of the National Health Service, the NIHR, or the UK Department of Health and Social Care.

## Author contributions

B.F.V. conceived the idea, designed, performed, and interpreted most of the experiments, and wrote the paper. S.N. performed and analyzed some Western blots, immunofluorescent staining experiments, and confocal microscopy. G.R.M. aided with some MN differentiations and the maintenance of cultures, and performed and analyzed some confocal microscopy. L.F. performed and analyzed some immunofluorescent staining experiments and confocal microscopy. E.C. performed some Western blots and performed and analyzed some immunofluorescent staining experiments. Y.X. aided with some MN differentiations, the maintenance of cultures and generation of samples, and performed RNAscope experiments. K.M.L.C. performed and interpreted the patch-clamping experiments. B.A. performed ELISA measurements. J.S. aided with some MN differentiations and the analysis of RNA sequencing data. A.K. aided with the maintenance of cultures and sample generation. A.C. aided with the design and interpretation of experiments. M.C. performed MSD ELISA measurements for DPR measurements. R.D. aided with some immunofluorescent staining experiments and confocal microscopy. A.M.I. supervised the MSD ELISA to measure DPR expression. R.W.M. supervised the patch-clamping experiments. E.G. aided with the design and interpretation of experiments. M.R.T. generated funding, interpreted experiments, and edited the paper. S.A.C. designed and interpreted experiments and edited the paper. K.T. generated funding, conceived the idea, designed and interpreted experiments, and edited the paper.

## Competing interests

The authors declare no competing interests.
