## [Peer Review File · Nature Communications]

REVIEWER COMMENTS

Reviewer #1 (Remarks to the Author):

The manuscript by Vahsen et al reports MMP9 as a crucial mediator of microglia to neuron toxicity in ALS due to a hexanucleotide repeat expansion (HRE) mutation in C9orf72. The Authors used induced pluripotent stem cell (iPSC)-derived microglia and motor neurons from 3 C9orf72 mutant patients (2 males and 1 female) and four control cell lines, including an isogenic control, with some variability in age. Notorious differences in microglia phenotype were only observed after microglia stimulation with lipopolysaccharide (LPS). The paper highlights the potential role of MMP9 in the DPP4 release in C9orf72 microglia cocultured with healthy motor neurons and suggests its use as a putative biomarker of early microglia dysfunction and the use of the DPP4 inhibitor as a therapeutic strategy. Thus, the study contributes to increase our understanding on microglia malfunction in the C9orf72 ALS disease and propose a new mediator in microglia-induced neurotoxicity upon inflammation. Nevertheless, there are several inconsistencies and weaknesses that the Authors should address with new data, while also improving the Discussion section with additional references and statements.

Major concerns about this paper are:

- The reduced number of cell lines that were investigated, which may limit the translational application of the findings;
- The incorrect designation of M0 and M1 microglia for what the Authors call unstimulated and LPS-primed microglia. This definition fell into disuse and these denominations must be eliminated from the manuscript. There are several papers addressing this issue, but the most relevant is from Paolicelli and all 2022 (doi: 10.1016/j.neuron.2022.10.020);
- The Authors use an in-vitro specific immune stimuli – the LPS, which activate differential metabolic programs and changes in cytokine expression. Though there are some papers referring to LPS entrance into the brain (<https://doi.org/10.1038/s41598-017-13302-6>) this is a controversial issue since most of the findings found LPS associated receptors mainly at blood-brain interface regions. In that way this type of immunostimulation is likely artificial and does not recapitulate the effects produced by neuroinflammation in the brain, where we may find different microglia activated states. The Authors should address this problem and at least reproduce some of their experiments with interferon-gamma stimulation or TNF-alpha+IL-1beta that better recapitulate the pathological inflammation in ALS and other neurodegenerative diseases;
- The Authors should avoid discussing their data on the Result section. In such cases, please use the Discussion section.
- Discussion section should be improved, and important missing references included, as indicated below.

Other relevant concerns:

- The Authors refer that microglia show typical microglial morphology (Figure 1). The quality of the images is poor. Please include images with larger number of cells together with insets. This ramified morphology can be found in different conditions, though is more usual in homeostatic conditions. Better to not refer to typical morphology. Besides, the transcriptomic analysis of microglia signature that the Authors previously published (doi: 10.1038/s41598-022-16896-8) deserve to be complemented with the functional assessment of the iPSC-derived microglia. Can the Authors include data on phagocytosis or migration properties when characterizing their first set of results in the manuscript in the absence and presence of LPS?
- Abbreviations should be the same throughout the text, figures, and supplementary material for readability. An example is the use of C9, pMGL and C9orf72 in the same Figure 1.
- Relatively to Figure 2, please indicate in the supplementary caption or in methods what were the parameters used in the GSEA platform to generate the pathway enrichment of LPS-stimulated vs. non stimulated. There is a reduced description of this part of the work in the manuscript. Besides, RNA seq data should be publicly available in a repository, and the link provided in the manuscript.
- Line 164: Although 62 DEGs were overlapping, it is important to know whether they inverted their regulation from non-stimulated to stimulated microglia. Those that showed to invert may be the most relevant, while the others may only be related to the C9orf72 genotype itself. Authors should discriminate differently expressed DEGs in non-stimulated vs. stimulated microglia and not to only focus on Calhm2.
- The Authors refer to significantly higher C9orf72 expression and relates it with endo-/lysosomal pathway. This aspect should be discussed in the Discussion Section and inclusion of papers reviewing this dysregulation included (e.g., endo-/lysosomal pathway), or some evaluations in the absence and presence of LPS, such as LC3II, p62, Lamp1, Rab7 and Rab9 should be evidenced for such involvement. Data will also reinforce the transcriptome analysis showing lysosome dysregulation, but without validation by RT-qPCR.
- Please briefly comment on negative enrichment in endoplasmic reticulum also in unstimulated mutant microglia, at least in the Discussion section.
- The Authors did not observe TDP-43 mislocalization and discuss this in the Result section. That should be better addressed in the Discussion Section and data from the paper by Lorenzini et al (doi: <https://doi.org/10.1101/2020.09.03.277459>), must be addressed. Such paper has many similarities with this paper, use iPSc-derived microglia and LPS and for sure deserve to be commented and related with Author's data.
- Another important point that is also missing is the inclusion of papers that addressed the disrupted autophagy of C9orf72-ALS iPSC-derived microglia also identifying MMP9, like the one by Banerjee et al (<https://doi.org/10.1101/2022.05.12.491675>).
- The Authors should describe the method to evaluate MMP9 activity and pro-MMP9. Was it by gelatin zymography? Please clarify in the Method section.

- How can the Authors justify the negative pathway enrichment in extracellular matrix (Fig. 2e) and the MMP9 increase? You should explore more such dichotomy in the discussion section, given the relevance of this.
- In their protocol, the Authors add microglia precursors and not mature cells added to MNs at a 1:1 ratio. The Authors need to explain in the MS the rationale to use such ratio and the microglia precursors, if they intend to reproduce an ALS pathological condition! Maturation of cells should be demonstrated before assessments. Some expected alterations only observed after LPS treatment may depend on that. Probably, because of that in Supp Fig. 5 the number of microglial cells largely surpassed the number of neurons, upon LPS what it is not physiologically relevant and may compromise the co-culture data. Authors need to address and justify these issues in the MS.
- It seems from the included data that the increase of MMP9 in the supernatant largely depend on LPS activation as observed in Figure 3, including the isogenic control. How far is related with LPS-specific induction? Authors need to address this hypothesis. Alternative stimulation would be interesting to validate or deny this hypothesis. Indeed, no statistical differences are indicated between “HC M0” and “C9 M0”. Please justify these points.
- Authors also reveal such increase in mixed microglia/MN cultures for the pro-MMP9 mainly. Usually, the designation of co-cultures is more applied when using inserts. Did the Authors alternatively tested such co-cultures to better follow the cell-to-cell communication of MMP9. That would also allow to assess the paracrine signaling consequences in each separate type of cells. Why did the Authors choose the mixed cultures? Please address this issue.
- When the Authors state that c9orf72 microglia are not toxic to motor neurons in co-culture can it be due to the defensive role of microglia precursor phenotypes on the healthy motor neurons, once they mature at the same time, what it is not occurring during neurodevelopment? Please clarify.
- Authors found a higher neuronal expression of the apoptotic marker CC3 upon LPS treatment in the motor neurons upon LPS-stimulated microglia by immunocytochemistry and in supernatants by ELISA. It is difficult to indicate enhanced apoptosis without cell demise after 14 days in mixed cultures. Co-culture with inserts would allow a better evaluation of increased levels of neuronal DPP4 enhancing the significance of the data. In the supernatants we can not be sure of its cellular origin. Moreover, since caspase-3 also functions as a regulatory molecule in neurogenesis and synaptic activity, such hypothesis should be alternatively considered (<https://www.nature.com/articles/cdd2009180/>). Once the Authors mention early stage apoptosis it would be interesting to reinforce the CC3 data with at least Annexin V assessment.
- Once MMP1 and MMP2 are also involved in DPP4 shedding data, evaluation of their activities would then benefit data validation and deserve to be assessed.

Minor comments:

- Line 17 but not unstimulated is redundant because means stimulated (do you mean non-stimulated?). Please rephrase.

- Line 172 “mutant microglia demonstrated positive enrichment of several terms associated with immune cell activation and cytokines/chemokines by GSEA” – please include relatively to stimulated healthy microglia.
- Line 121 at Supplementary Material – stored at 80°C: should be stored at -80°C.
- Figure 2 page 31, please explain the black symbols in panel a. Are the colors wrong?

Reviewer #2 (Remarks to the Author):

This paper is more substantial than analogous papers available using similar tools to address similar questions, so my reaction is positive. Importantly these authors demonstrate both a non-cell autonomous effect on neurons and attribute a MMP-9 -dependent mechanism. MMP9 has indeed been proposed before as a salient target in ALS, but this study importantly puts this into an intercellular cellular and human experimental context. I would be keen to see a revised version of this paper for further consideration.

Concerns to address:

- 1) The neuron cultures show a ball and chain type of pattern caused by clumping of the cell bodies, which makes accurate quantification difficult on immunofluorescence – can the authors comment on how they dealt with this? Given this appearance of the cultures, authors need to summarise their quality checks for each line, each induction and comment on the reproducibility of differentiation between lines / inductions. Also per field data are plotted rather than averaging the fields per line/differentiation, which is not ideal and should be corrected.
- 2) It is very difficult in co-cultures to achieve an ideal ratio of microglia to neurons. Here the authors have plated these 1:1 which is not particularly physiological. Duration of these experiments is then 14 days, which may partly account for large variation noted cell count data in the last supplementary figure for example. Such differences in the number of microglia, even if not significant, may drive some of the phenotypes observed in the C9orf72 microglia. Can the authors try to address this concern experimentally?
- 3) Do C9 microglia show repeat foci? This has previously been shown in iPSC-derived neurons but would be important to investigate here

4) Mutant microglia on mutant neurons as a co-culture paradigm seems to have been overlooked by the authors as beyond the scope of the study but it seems a rather integral part in my view

5) I would see the lack of isogenic rescue of haploinsufficiency as a concern rather than the opportunity seen by the authors – please can you comment on why this is the case? Is the correction accurate for example

6) RNA-seq - library preparation is not clear (polyA or ribo-zero)? Was this stranded or unstranded? What were the read lengths? What was the average read depth per sample?

7) Figure 2A: the PCA separated in PC2 by LPS treatment and PC6 by ALS vs CTRL. It is unusual to show a PC2 vs PC6. It would be worth examining the genes driving PC6 separation. Also, in addition to a PCA, it would be helpful if the authors show a volcano or MA plot showing the ALS vs CTRL differential gene expression for untreated and treated separately and annotate differentially expressed genes.

8) Figure 2B: the overlap of differentially expressed genes should indicate the direction of change. It is unclear what proportion of the overlap is upregulated and what is downregulated in ALS / CTRL.

9) To better show the relationship between M0 and M1 in ALS vs CTRL, can the authors show in a scatter plot of the correlation of transcriptome-wide changes using the log2foldchange or test statistic in ALS/CTRL for M0 with ALS/CTRL M1?

10) Technical repeats in the RNAseq should really be merged as they are artificially inflating their sample size with this approach.

Reviewer #3 (Remarks to the Author):

This is an interesting paper from an established team, suggesting a novel role for MMP9, especially within the context of inflammation in C9orf72-ALS.

c9orf72-ALS and inflammation are important topics as c9orf72 HRE account for a large population of ALS and inflammation is one of the converging paths when it comes to neurodegeneration. Therefore, this manuscript hits two important and relevant topics.

I have the following remarks/edits/comments to make:

1) Authors use 3 different C9orf72 ALS patient lines, 3 healthy controls, and 3 isogenic control lines. Do the patient lines carry the same genetic problem, ie. same number of HRE expansion? How "healthy" are the healthy controls? Are they non-ALS patients, other disease patients, how about their WGS, does that show any other potential disease, any other genetic abnormalities? I think a better identification and clarification is required on the cell lines. Especially now that there are only n=3 (relatively small number), it would be nice to know more about the origin of these cells.

2) It is interesting that the diseased iPSCs display microgliosis only after and only if stimulated by LPS. This is very interesting and strange. So the disease state, even though they have the DPRs, is not sufficient for them to activate microgliosis? As far as I know there are other disease iPSCs that are reported to have enhanced microgliosis. How different are these cells from the previously published?

3) How pure are the differentiated cell lines? For example, when cells differentiate into microglia, what percent of the cells in the plate are indeed microglia? This is especially important for co-culture experiments. When the motor neuron cultures, (I assume they are CHAT+ spinal motor neurons, correct?, the spinal motor neuron identity of these neurons need to be specific in the text) are grown together with the microglia cultures, what percent of the cells are actually spinal motor neurons? All cellular analyses are performed after the experiment is completed, correct? Western blot may show relative protein levels, but that also needs to be normalized or corrected based on to the total number of cells of that type, not total number of cells including all types. That is why I think a detailed immunocytochemical analyses is required to reveal which cell is which, what is their percent distribution on the plate and what is the cell-cell interaction of differentiated spinal motor neurons with activated or inactivated microglia.

3) Not knowing what percent of all cells in the plate differentiate to microglia or any given cell-type is an important hurdle also for the RNA-Seq experiments because we will never know if these data were obtained from "pure" microglia or pure "motor neuron" cells. Maybe a FACS purification based on forward and side scatter characteristics of microglia may be utilized. A small experiment may be performed to investigate potential differences in the data obtained while using mixed cells versus using purified cells.

4) Authors have claim that the culture time might not be long enough to detect cellular degeneration or clearance, but that they detect increased expression of apoptotic markers and they use this as an outcome measure to assess toxicity. They were treated for 10 days with LPS. This is a very long time for LPS treatment. Maybe they should do one more extended time experiment, maybe treat with longer time to further confirm that increased expression of CC3, does indeed lead to neuronal degeneration. I think there is more and stronger evidence required than just increased CC3 expression.

5) MMP9 is a protein that comes in the pro-MMP9 form and it is cleaved to become active and there are specific enzymes that cleave this protein. I think the authors should look into the presence and the expression of these proteins/enzymes? without these proteins MMP9 will not be active and pro-MMP9 is not functionally active as the MMP9. They mention MMP9 inhibitor. What is the mode of action of this compound? Does it inhibit the enzyme that blocks the conversion of pro-MMP9 to MMP9?

Not being able to observe microglial activation with just the disease state and the fact that it needs to be promoted and pushed with LPS stimulation, and the fact that the outcome measure depends on the detection of increased CC3 expression weakens the enthusiasm on a very important and intriguing finding.

I hope the authors will perform additional experiments to further support the important claims that they make. Overall, I think this is an important paper and results need to be further enhanced with proper controls and additional experimentation.

Minor:

1) The text in the figures are very very small. None of the y-axis of the bar graphs can be read.

2) Maybe Fig 4 can be divided into two?

3) Figure 5g can be enlarged, there is space in the figure.

Thank you.

Rebuttal Letter

REVIEWER COMMENTS

Reviewer #1 (Remarks to the Author):

The manuscript by Vahsen et al reports MMP9 as a crucial mediator of microglia to neuron toxicity in ALS due to a hexanucleotide repeat expansion (HRE) mutation in C9orf72. The Authors used induced pluripotent stem cell (iPSC)-derived microglia and motor neurons from 3 C9orf72 mutant patients (2 males and 1 female) and four control cell lines, including an isogenic control, with some variability in age. Notorious differences in microglia phenotype were only observed after microglia stimulation with lipopolysaccharide (LPS). The paper highlights the potential role of MMP9 in the DPP4 release in C9orf72 microglia cocultured with healthy motor neurons and suggests its use as a putative biomarker of early microglia dysfunction and the use of the DPP4 inhibitor as a therapeutic strategy. Thus, the study contributes to increase our understanding on microglia malfunction in the C9orf72 ALS disease and propose a new mediator in microglia-induced neurotoxicity upon inflammation. Nevertheless, there are several inconsistencies and weaknesses that the Authors should address with new data, while also improving the Discussion section with additional references and statements.

Reply: We thank the reviewer for their detailed comments and suggestions, which have improved our manuscript. We provide a detailed response addressing the reviewer's comments below.

Major concerns about this paper are:

- The reduced number of cell lines that were investigated, which may limit the translational application of the findings;

Reply: Our C9orf72-ALS lines have been validated in studies in several previous publications and are derived from clinically well-characterized C9orf72-ALS patients. Importantly, the key C9orf72-ALS microglial phenotypes we describe are consistent across comparisons between healthy and isogenic control microglia. We are therefore confident that our findings, while clearly restricted to the *in vitro* domain at present, are relevant to C9orf72-ALS. We are currently planning follow-up studies using a range of other ALS mutations (TDP-43 etc) to shed more light on the broader translational applicability of our findings.

- The incorrect designation of M0 and M1 microglia for what the Authors call unstimulated and LPS-primed microglia. This definition fell into disuse and these denominations must be eliminated from the manuscript. There are several papers addressing this issue, but the most relevant is from Paolicelli and all 2022 (doi: 10.1016/j.neuron.2022.10.020);

Reply: We completely agree with the reviewer and have removed all references to the terms 'M0' and 'M1' from the manuscript. Instead, we now mention the specific stimuli used (LPS or TNF/IL1B).

- The Authors use an in-vitro specific immune stimuli – the LPS, which activate differential metabolic programs and changes in cytokine expression. Though there are some papers referring to LPS entrance into the brain (<https://doi.org/10.1038/s41598-017-13302-6>) this a controversial issue once most of the findings found LPS associated receptors mainly at blood-brain interface regions. In that way this type of immunostimulation is likely artificial and do not recapitulate the effects produced by neuroinflammation in the brain, where we may find different microglia activated states. The Authors should address this problem and at least reproduce some of their experiments with interferon-gamma stimulation or TNF-alpha+IL-1beta that better recapitulate the pathological inflammation in ALS and other neurodegenerative diseases;

Reply: LPS is a standard way of modelling microglial activity in vitro¹. But to address the reviewer's point, we have performed some additional experiments with TNF/IL1B treatment. Although slightly

less pronounced than after LPS stimulation, we found increased MMP9 expression by Western blot and ELISA in C9orf72 mutant microglia compared with controls after TNF/IL1B priming (Supp. Fig. 5f).

- The Authors should avoid discussing their data on the Result section. In such cases, please use the Discussion section.

Reply: We have removed some statements from the Results section and expanded our Discussion section.

- Discussion section should be improved, and important missing references included, as indicated below.

Reply: We now discuss the papers suggested by the reviewer in the discussion.

Other relevant concerns:

- The Authors refer that microglia show typical microglial morphology (Figure 1). The quality of the images is poor. Please include images with larger number of cells together with insets. This ramified morphology can be found in different conditions, though is more usual in homeostatic conditions. Better to not refer to typical morphology. Besides, the transcriptomic analysis of microglia signature that the Authors previously published (doi: 10.1038/s41598-022-16896-8) deserve to be complemented with the functional assessment of the iPSC-derived microglia. Can the Authors include data on phagocytosis or migration properties when characterizing their first set of results in the manuscript in the absence and presence of LPS?

Reply: We have added new images to Figure 1 including insets (Fig. 1b). We have removed the term 'typical' from the manuscript and have confirmed that microglia from all iPSC lines used in this study are phagocytically active, which is increased after LPS treatment (Supp. Fig. 1g/h).

- Abbreviations should be the same throughout the text, figures, and supplementary material for readability. An example is the use of C9, pMGL and C9orf72 in the same Figure 1.

Reply: We apologise for this oversight and have tried to unify the abbreviations used in the manuscript. If space permits, we use the whole gene name (C9orf72), otherwise we use 'C9'.

- Relatively to Figure 2, please indicate in the supplementary caption or in methods what were the parameters used in the GSEA platform to generate the pathway enrichment of LPS-stimulated vs. non stimulated. There is a reduced description of this part of the work in the manuscript. Besides, RNA seq data should be publicly available in a repository, and the link provided in the manuscript.

Reply: We have added more information to the methods section (pp. 27/28). We had already uploaded the RNA seq data to the GEO database and provided the accession in the manuscript (p. 31). The dataset is currently accessible to reviewers through a token provided to the journal, and the dataset will be made publicly available after publication. To review GEO accession GSE217625, please go to <https://www.ncbi.nlm.nih.gov/geo/query/acc.cgi?acc=GSE217625>, and enter token ehofkogufgrnlwl into the box.

- Line 164: Although 62 DEGs were overlapping, it is important to know whether they inverted their regulation from non-stimulated to stimulated microglia. Those that showed to invert may be the most relevant, while the others may only be related to the C9orf72 genotype itself. Authors should discriminate differently expressed DEGs in non-stimulated vs. stimulated microglia and not to only focus on Calhm2.

Reply: We now provide a Venn diagram in the main figure (Fig. 2b) showing the total overlap as well as two different Venn diagrams in the supplement (Supp. Fig. 3d), showing the overlap between up- and downregulated genes in both comparisons separately. We also provide volcano plots showing

the DEGs for the different comparisons (unstimulated and stimulated; Supp. Fig. 3f/g, Supp. Fig. 4e/f). Please note that we have re-analysed the data merging the different differentiations into one sample per line as requested by reviewer 3 (Fig. 2, Supp. Fig. 3). With this analysis, *CXCL1* and *CXCL6* are DEGs upregulated in *C9orf72* mutant microglia after LPS priming, which we have added to the main figure 2 (Fig. 2d). We believe that *Calhm2* (as well as the other DEGs previously identified) are also relevant and therefore additionally show volcano plots analysing the different differentiations as separate data points in the supplement (as done previously) (Supp. Fig. 4e/f).

- The Authors refer to significantly higher *C9orf72* expression and relates it with endo-/lysosomal pathway. This aspect should be discussed in the Discussion Section and inclusion of papers reviewing this dysregulation included (e.g., endo-/lysosomal pathway), or some evaluations in the absence and presence of LPS, such as LC3II, p62, Lamp1, Rab7 and Rab9 should be evidenced for such involvement. Data will also reinforce the transcriptome analysis showing lysosome dysregulation, but without validation by RT-qPCR.

Reply: We have added a short paragraph to the discussion (p. 18). We agree that a more detailed experimental investigation of *C9orf72* in the endo-lysosomal pathway in microglia would be interesting but have opted against performing further experiments to focus on further corroborating the non-cell-autonomous microglial toxicity. We will endeavour to address this in future studies.

- Please briefly comment on negative enrichment in endoplasmic reticulum also in unstimulated mutant microglia, at least in the Discussion section.

Reply: We have added this to the results section (p. 8) and the summary schematic in Fig. 2f and also mention it in the discussion (p. 18).

- The Authors did not observe TDP-43 mislocalization and discuss this in the Result section. That should be better addressed in the Discussion Section and data from the paper by Lorenzini et al (doi: <https://doi.org/10.1101/2020.09.03.277459>), must be addressed. Such paper has many similarities with this paper, use iPSC-derived microglia and LPS and for sure deserve to be commented and related with Author's data.

Reply: We have added a short statement on this to the discussion section and compare our findings with the study by Lorenzini et al. (p. 17). As this is not the main focus of our paper and to comply with space constraints, we have not entered into an extended discussion (nb. the analysis of TDP-43 is now in Supp. Fig. 2f to free up space for the analysis of RNA foci).

- Another important point that is also missing is the inclusion of papers that addressed the disrupted autophagy of *C9orf72*-ALS iPSC-derived microglia also identifying MMP9, like the one by Banerjee et al (<https://doi.org/10.1101/2022.05.12.491675>).

Reply: We have added this to the discussion section (pp. 16-21).

- The Authors should describe the method to evaluate MMP9 activity and pro-MMP9. Was it by gelatin zymography? Please clarify in the Method section.

Reply: We have used Western blot to discriminate pro-MMP9 and cleaved (active) form of MMP9 in cell lysates by molecular weight. Total MMP9 activity in the supernatant was detected by a fluorometric assay detecting MMP9 activity by cleavage of a peptide linker between the fluorophore and a quencher molecule after activation of all MMP9 forms using APMA. We have made this clearer in the respective figure legends (Fig. 3b/d).

- How can the Authors justify the negative pathway enrichment in extracellular matrix (Fig. 2e) and the MMP9 increase? You should explore more such dichotomy in the discussion section, given the relevance of this.

Reply: After re-analysing our data using the merging of differentiations requested by reviewer 3, ECM-associated terms were not amongst the top dysregulated pathways. We have therefore decided to remove this statement from the manuscript and summary schematic (Fig. 2f).

- In their protocol, the Authors add microglia precursors and not mature cells added to MNs at a 1:1 ratio. The Authors need to explain in the MS the rationale to use such ratio and the microglia precursors, if they intend to reproduce an ALS pathological condition! Maturation of cells should be demonstrated before assessments. Some expected alterations only observed after LPS treatment may depend on that. Probably, because of that in Supp Fig. 5 the number of microglial cells largely surpassed the number of neurons, upon LPS what it is not physiologically relevant and may compromise the co-culture data. Authors need to address and justify these issues in the MS.

Reply: In neurodevelopment, yolk-sac derived microglial precursors migrate into the brain and spinal cord and then terminally mature into microglia in situ. To mimic this process, we have added microglial precursors to young MNs and then allowed them to mature together. We have added new data showing high expression of the spinal MN marker ChAT in MNs in co-culture, confirming MN maturation (Supp. Fig 6c, Supp. Fig. 8e). The reason why we used a 1:1 ratio is because the overall neuron:glia ratio in the CNS is thought to be close to 1:1 determined by a recent meta-analysis². We are not aware of studies detailing a specific motor neuron:microglia ratio in the human spinal cord and therefore used a 1:1 ratio as a reference point for our model system. We have added a brief explanation to the manuscript (p. 10). We also thank the reviewer for pointing out the inconsistency between the MN and microglia counts, which was due to different image sizes used for the counting of MNs and microglia. We have now rectified this (Supp. Fig. 6a/b, Supp. Fig. 8a/b) and included a quantification of the microglia:MN ratio (Fig. 4b, Fig. 5b), showing that the microglia:MN ratio is ~1:1 in unstimulated co-cultures and approximately 1:2 after stimulation with LPS.

- It seems from the included data that the increase of MMP9 in the supernatant largely depend on LPS activation as observed in Figure 3, including the isogenic control. How far is related with LPS-specific induction? Authors need to address this hypothesis. Alternative stimulation would be interesting to validate or deny this hypothesis. Indeed, no statistical differences are indicated between “HC M0” and “C9 M0”. Please justify these points.

Reply: We already observe a trend to higher MMP9 expression in C9 conditions vs unstimulated HC, but LPS priming further increases differences in MMP9 expression levels. We have now added data to the supplement evaluating MMP9 expression after priming with TNF/IL1B (Supp. Fig. 5f), as mentioned above. Although slightly less pronounced than after LPS treatment, we show increased MMP9 expression by Western blot and ELISA also after TNF/IL1B priming between C9orf72 mutant microglia and controls.

- Authors also reveal such increase in mixed microglia/MN cultures for the pro-MMP9 mainly. Usually, the designation of co-cultures is more applied when using inserts. Did the Authors alternatively tested such co-cultures to better follow the cell-to-cell communication of MMP9. That would also allow to assess the paracrine signaling consequences in each separate type of cells. Why did the Authors choose the mixed cultures? Please address this issue.

Reply: We chose mixed microglia/MN co-cultures to 1) allow both cell types to mature together and 2) to model contact-dependent and contact-independent cellular crosstalk in parallel. Microglia maturation in vitro is supported by co-culture with neurons, as described in our previous publications^{3,4}, and inserts/transwells would abolish this cell-cell interaction. We have now additionally performed a new co-culture experiment with microglia cultured in transwells, demonstrating relevant non-cell-autonomous toxicity of C9orf72 mutant microglia in the absence of direct contact, suggesting the neurotoxic effects of C9orf72 microglia are partly driven by soluble mechanisms (Supp. Fig. 8f). Similarly, we now show that direct treatment with human recombinant

MMP9 reduces MN viability, which is rescued by concomitant treatment with MMP9 inhibitor (Supp. Fig. 9e).

- When the Authors state that C9orf72 microglia are not toxic to motor neurons in co-culture can it be due to the defensive role of microglia precursor phenotypes on the healthy motor neurons, once they mature at the same time, what it is not occurring during neurodevelopment? Please clarify.

Reply: As detailed above, our model mimics neurodevelopment where neurons and microglia also mature together. The addition of microglia in the cultures generally supports neuronal health and, our data suggest that non-stimulated co-cultures are in a homeostatic state. Stimulation seems to be required to uncover non-cell-autonomous microglial neurotoxicity.

- Authors found a higher neuronal expression of the apoptotic marker CC3 upon LPS treatment in the motor neurons upon LPS-stimulated microglia by immunocytochemistry and in supernatants by ELISA. It is difficult to indicate enhanced apoptosis without cell demise after 14 days in mixed cultures. Co-culture with inserts would allow a better evaluation of increased levels of neuronal DPP4 enhancing the significance of the data. In the supernatants we can not be sure of its cellular origin. Moreover, since caspase-3 also functions as a regulatory molecule in neurogenesis and synaptic activity, such hypothesis should be alternatively considered (<https://www.nature.com/articles/cdd2009180/>). Once the Authors mention early stage apoptosis it would be interesting to reinforce the CC3 data with at least Annexin V assessment.

Reply: As detailed above, we have now added a co-culture experiment with inserts showing reduced neuronal viability in co-culture with C9orf72 microglia (Supp. Fig 8f). Furthermore, we have now added an experiment with long-term LPS stimulation, as suggested by reviewer 3, confirming increased CC3 expression and overt neurotoxicity of C9orf72 mutant microglia (Fig. 5i-k). We therefore consider that the increased expression of CC3 is indeed reflective of an increased activation of neuronal cell death pathways. We agree that an evaluation of Annexin V would be interesting but have decided to lay the focus of the revision experiments on evaluating overt neurodegeneration after long-term LPS exposure.

- Once MMP1 and MMP2 are also involved in DPP4 shedding data, evaluation of their activities would then benefit data validation and deserve to be assessed.

Reply: We had previously assessed both MMP1 and MMP2 in supernatants from in C9orf72 mutant microglia monocultures by a protease/protease inhibitor array (Supp. Fig 5g), showing a relative decrease in MMP1 release and an increase in MMP2 secretion. Our RNA sequencing data showed no differences for *MMP1* and a trend to decreased *MMP2* expression in C9orf72 mutant microglia. (Supp. Fig. 5h). We therefore deemed MMP1 less relevant. As the relative MMP2 increase in the supernatant was comparatively high, we have attempted to perform a WB for MMP2 to assess its cellular expression in C9orf72 mutant microglia at protein level but have not been able to achieve a reliable signal. We therefore decided not to further explore MMP2.

Minor comments:

- Line 17 but not unstimulated is redundant because means stimulated (do you mean non-stimulated?). Please rephrase.

Reply: We have rephrased this sentence.

- Line 172 “mutant microglia demonstrated positive enrichment of several terms associated with immune cell activation and cytokines/chemokines by GSEA” – please include relatively to stimulated healthy microglia.

Reply: We have added this phrase to the sentence.

- Line 121 at Supplementary Material – stored at 80°C: should be stored at -80°C.

Reply: We thank the reviewer for spotting this typo.

- Figure 2 page 31, please explain the black symbols in panel a. Are the colors wrong?

Reply: The circular symbols are unstimulated microglia ('M0', now renamed to -LPS), while the triangular shaped symbols are LPS-primed ('M1', now renamed to +LPS).

Reviewer #2 (Remarks to the Author):

This paper is more substantial than analogous papers available using similar tools to address similar questions, so my reaction is positive. Importantly these authors demonstrate both a non-cell autonomous effect on neurons and attribute a MMP-9 -dependent mechanism. MMP9 has indeed been proposed before as a salient target in ALS, but this study importantly puts this into an intercellular cellular and human experimental context. I would be keen to see a revised version of this paper for further consideration.

Reply: We thank the reviewer for their detailed comments and suggestions, which have improved our manuscript. We provide a detailed response addressing the reviewer's comments below.

Concerns to address:

- 1) The neuron cultures show a ball and chain type of pattern caused by clumping of the cell bodies, which makes accurate quantification difficult on immunofluorescence – can the authors comment on how they dealt with this? Given this appearance of the cultures, authors need to summarise their quality checks for each line, each induction and comment on the reproducibility of differentiation between lines / inductions. Also per field data are plotted rather than averaging the fields per line/differentiation, which is not ideal and should be corrected.

Reply: We had previously counted the number of cells positive for CC3, which we agree is challenging with cell clusters. We have therefore now used a macro-based quantification in Fiji where the MN area positive for CC3 is automatically quantified. This quantification replicates our previous finding of increased CC3 expression in MNs in co-culture with C9orf72 mutant microglia after manual and blinded quantification and is faster and unbiased (Fig. 5f). As for the quality checks, we had selected a healthy control line (HC-2b) to differentiate MNs for the co-culture experiments that very reproducibly generates MNs with high ChAT expression (images and quantification now added to the manuscript, Supp. Fig. 6c, Supp. Fig. 8e). As we wanted to focus on the non-cell-autonomous effects of microglia on motor neurons and avoid potentially confounding effects of slightly varying differentiation efficiencies into MNs between different iPSC lines, we had opted to use the same healthy control line to generate MNs for co-cultures, with multiple independent (co-culture) differentiations for this line, similar to the approaches by Frakes et al.⁵ and Birger et al.⁶ We have now made this clearer in the methods part of the manuscript (p. 25) and legend for Supplementary Table 1. Finally, we agree with the reviewer and have averaged the field data per differentiation for all relevant graphs and have increased the number of differentiations for some experiments.

- 2) It is very difficult in co-cultures to achieve an ideal ratio of microglia to neurons. Here the authors have plated these 1:1 which is not particularly physiological. Duration of these experiments is then 14 days, which may partly account for large variation noted cell count data in the last supplementary figure for example. Such differences in the number of microglia, even if not significant, may drive some of the phenotypes observed in the C9orf72 microglia. Can the authors try to address this concern experimentally?

Reply: As detailed above, the reason why we used a 1:1 ratio is because the overall neuron:glia ratio in the CNS is thought to be close to 1:1 determined by a recent meta-analysis². We are not aware of studies detailing a specific MN:microglia ratio in the human spinal cord and therefore used a 1:1 ratio as a reference point for our model system. Regarding the cell count data, as pointed out above,

we had accidentally used differently sized images, which we have now rectified (Supp. Fig. 6a/b, Supp. Fig. 8a/b). While there is some variation in the number of microglia, this is true for both control and disease microglia. As we do not see differences in the cell count between C9orf72 mutant microglia and controls, we are of the view that the phenotypes observed in C9orf72 microglia are intrinsic to the disease condition and not connected with slightly varying microglia numbers.

3) Do C9 microglia show repeat foci? This has previously been shown in iPSC-derived neurons but would be important to investigate here

Reply: We agree that this is an important question and have performed an additional experiment using RNAscope to detect RNA foci (Fig. 1). We demonstrate that both sense and anti-sense RNA foci are present in C9orf72 HRE mutant iPSC microglia, with more microglia positive for anti-sense than sense foci (Fig. 1h, Supp. Fig. 1e).

4) Mutant microglia on mutant neurons as a co-culture paradigm seems to have been overlooked by the authors as beyond the scope of the study but it seems a rather integral part in my view

Reply: We agree with the reviewer that mutant microglia on mutant neurons are a very important experimental paradigm. For this study, however, we decided to focus on the non-cell-autonomous consequences of microglia on neurons; focusing on the effects of microglia on healthy control motor neurons allows for a more reproducible experimental approach by reducing additional variance caused by different MN lines (irrespective of the genotype). For experiments comparing co-cultures with mutant and control motor neurons, all experiments will need to be conducted at the same time to limit batch effects and we would therefore have to repeat all co-culture experiments from this manuscript, with multiple control and disease lines differentiated into MNs, to address this question properly. We therefore feel that this question is best addressed in a major follow up project and is beyond the scope of the current study.

5) I would see the lack of isogenic rescue of haploinsufficiency as a concern rather than the opportunity seen by the authors – please can you comment on why this is the case? Is the correction accurate for example

Reply: Neither RNA foci nor DPR are expressed after correction of the hexanucleotide repeat expansion in the isogenic line (Fig. 1), indicating accurate correction of the repeat. It might be possible that the lack of isogenic rescue of haploinsufficiency is due to epigenetic effects of the mutation that have not been reversed after expansion removal or, alternatively, due to selection of a clone with equal C9orf72 expression during the CRISPR correction. More isogenic lines from different donor lines, which we are in the process of generating, will be able to answer this question in future studies.

6) RNA-seq - library preparation is not clear (polyA or ribo-zero)? Was this stranded or unstranded? What were the read lengths? What was the average read depth per sample?

Reply: We have added this information to the methods (pp. 27/28).

7) Figure 2A: the PCA separated in PC2 by LPS treatment and PC6 by ALS vs CTRL. It is unusual to show a PC2 vs PC6. It would be worth examining the genes driving PC6 separation. Also, in addition to a PCA, it would be helpful if the authors show a volcano or MA plot showing the ALS vs CTRL differential gene expression for untreated and treated separately and annotate differentially expressed genes.

Reply: We have added a short statement on loadings for PC6, as, interestingly, the biomarker candidate CHIT1 is the second most highly contributing gene for PC6 loading (p. 7). We now also provide volcano plots for both comparisons in the supplement (Supp. Fig. 3f/g, Supp. Fig. 4e/f).

8) Figure 2B: the overlap of differentially expressed genes should indicate the direction of change. It is unclear what proportion of the overlap is upregulated and what is downregulated in ALS / CTRL.

Reply: We have added Venn diagrams for the up-regulated and down-regulated genes to the supplement (Supp. Fig. 3d) and have also coloured the DEGs in the different comparisons in the scatter plot suggested by the reviewer below (Supp. Fig. 3e).

9) To better show the relationship between M0 and M1 in ALS vs CTRL, can the authors show in a scatter plot of the correlation of transcriptome-wide changes using the log2foldchange or test statistic in ALS/CTRL for M0 with ALS/CTRL M1?

Reply: We have added a scatter plot comparing the log2fc in both comparisons to the supplement (Supp. Fig. 3e).

10) Technical repeats in the RNAseq should really be merged as they are artificially inflating their sample size with this approach.

Reply: We are unsure if the reviewer refers to different sequencing runs or different independent differentiations for the same sample when they mention 'technical replicates'. We had already merged the two different sequencing runs for the same samples in our previous analysis. We have now additionally re-analysed our data merging the three different differentiations per sample (effectively comparing n=3 lines/datapoints for C9 microglia vs n=3 lines/datapoints for HC microglia). Importantly, the central messages of the RNA sequencing are still retained (Fig. 2, Supp. Fig. 3, Supp. Fig. 4). The individual DEGs differ slightly between this analysis and our previous analysis. In our view, DEGs from both analyses are relevant and valuable, and while we now primarily focus on data with the different differentiations merged, we have decided to show volcano plots for both analyses in the manuscript (new analysis: Supp. Fig. 3f/g; previous analysis: Supp. Fig. 4e/f).

Reviewer #3 (Remarks to the Author):

This is an interesting paper from an established team, suggesting a novel role for MMP9, especially within the context of inflammation in C9orf72-ALS.

c9orf72-ALS and inflammation are important topics as c9orf72 HRE account for a large population of ALS and inflammation is one of the converging paths when it comes to neurodegeneration. Therefore, this manuscript hits two important and relevant topics.

Reply: We thank the reviewer for their detailed comments and suggestions, which have improved our manuscript. We provide a detailed response addressing the reviewer's comments below.

I have the following remarks/edits/comments to make:

1) Authors use 3 different C9orf72 ALS patient lines, 3 healthy controls, and 3 isogenic control lines. Do the patient lines carry the same genetic problem, ie. same number of HRE expansion? How "healthy" are the healthy controls? Are they non-ALS patients, other disease patients, how about their WGS, does that show any other potential disease, any other genetic abnormalities? I think a better identification and clarification is required on the cell lines. Especially now that there are only n=3 (relatively small number), it would be nice to know more about the origin of these cells.

Reply: We have provided an estimate of the HRE expansion size in the supplement for the different disease lines (Supplementary Table 1). The healthy controls are age-matched healthy, unrelated, individuals with no known neurological disease. They have not been subjected to WGS. The C9orf72 lines come from ALS cases seen in the Oxford clinic. In all cases the repeat size was given by clinical DNA laboratory testing as >500 repeats, so well within the established range for pathogenicity.

2) It is interesting that the diseased iPSCs display microgliosis only after and only if stimulated by LPS. This is very interesting and strange. So the disease state, even though they have the DPRs, is not

sufficient for them to activate microgliosis? As far as I know there are other disease iPSCs that are reported to have enhanced microgliosis. How different are these cells from the previously published?

Reply: It is clear from a range of pathological studies and in vitro model systems that DPRs can be tolerated in neurons for long periods, suggesting they are not a sufficient condition for the disease state, and other co-factors are required. ALS, including that due to C9orf72, is well established to be a 'multiple hit' process. Indeed, our data suggest that unstimulated C9orf72 microglia only have a mild intrinsic phenotype, which requires stimulation to become apparent. This observation is in line with the very recent publications on C9orf72 mutant microglia by Banerjee et al.⁷ and Lorenzini et al.⁸ We have added a discussion of these articles to our Discussion section (pp. 16-21).

3) How pure are the differentiated cell lines? For example, when cells differentiate into microglia, what percent of the cells in the plate are indeed microglia? This is especially important for co-culture experiments. When the motor neuron cultures, (I assume they are CHAT+ spinal motor neurons, correct?, the spinal motor neuron identity of these neurons need to be specific in the text) are grown together with the microglia cultures, what percent of the cells are actually spinal motor neurons? All cellular analyses are performed after the experiment is completed, correct? Western blot may show relative protein levels, but that also needs to be normalized or corrected based on to the total number of cells of that type, not total number of cells including all types. That is why I think a detailed immunocytochemical analyses is required to reveal which cell is which, what is their percent distribution on the plate and what is the cell-cell interaction of differentiated spinal motor neurons with activated or inactivated microglia.

Reply: The microglia differentiation protocol generates a high proportion of IBA1 and TMEM119-positive cells; we have now added a quantification of these markers to the supplementary information (Supp. Fig. 1a-c). We have also clarified the spinal identity of the motor neurons in the manuscript. As the reviewer suggested, we now also provide a quantification of ChAT expression in unstimulated and LPS-stimulated co-cultures (Supp. Fig. 6c, Supp. Fig. 8e) as well as an immunocytochemical quantification of the microglia:MN ratio (Fig. 4b, Fig. 5b).

4) Not knowing what percent of all cells in the plate differentiate to microglia or any given cell-type is an important hurdle also for the RNA-Seq experiments because we will never know if these data were obtained from "pure" microglia or pure "motor neuron" cells. Maybe a FACS purification based on forward and side scatter characteristics of microglia may be utilized. A small experiment may be performed to investigate potential differences in the data obtained while using mixed cells versus using purified cells.

Reply: As noted above, the purity of the microglia differentiation is very high, and therefore the RNA seq experiment will have been done on a fairly pure microglial population. We have also previously shown⁴ that the transcriptome of our iPSC microglia quite closely resembles bona fide microglia. Therefore, we feel that the FACS experiment suggested by the reviewer would not substantially add to this data.

5) Authors have claim that the culture time might not be long enough to detect cellular degeneration or clearance, but that they detect increased expression of apoptotic markers and they use this as an outcome measure to assess toxicity. They were treated for 10 days with LPS. This is a very long time for LPS treatment. Maybe they should do one more extended time experiment, maybe treat with longer time to further confirm that increased expression of CC3, does indeed lead to neuronal degeneration. I think there is more and stronger evidence required than just increased CC3 expression.

Reply: We agree with the reviewer and have performed a new experiment with prolonged treatment with LPS, as suggested (Fig. 5i-k). We confirm increased expression of CC3 and show a reduction in

the relative MN number after prolonged LPS treatment in co-cultures with C9orf72 mutant microglia compared with controls, indicating overt neurotoxicity.

6) MMP9 is a protein that comes in the pro-MMP9 form and it is cleaved to become active and there are specific enzymes that cleave this protein. I think the authors should look into the presence and the expression of these proteins/enzymes? without these proteins MMP9 will not be active and pro-MMP9 is not functionally active as the MMP9. They mention MMP9 inhibitor. What is the mode of action of this compound? Does it inhibit the enzyme that blocks the conversion of pro-MMP9 to MMP9?

Reply: We agree that it would be interesting to evaluate the upstream signaling leading to MMP9 upregulation. We had already assessed the release of some more proteases from C9orf72 mutant microglia, some of which can activate MMP9, using a protease array (Supp. Fig. 5g/i), but believe that further characterisation is beyond the scope of this paper. Furthermore, Western blot allows for the distinction of pro-MMP9 and active MMP9 by molecular weight, and we have demonstrated that active MMP9 is significantly increased in C9orf72 mutant microglia (Fig. 3d). The MMP9 inhibitor used in this paper (MMP9 inhibitor 1) is a cell-permeable inhibitor and inhibits MMP9 activity⁹.

Not being able to observe microglial activation with just the disease state and the fact that it needs to be promoted and pushed with LPS stimulation, and the fact that the outcome measure depends on the detection of increased CC3 expression weakens the enthusiasm on a very important and intriguing finding.

I hope the authors will perform additional experiments to further support the important claims that they make. Overall, I think this is an important paper and results need to be further enhanced with proper controls and additional experimentation.

Reply: We thank the reviewer for the positive feedback and hope that we have now provided sufficient further data to substantiate our conclusions.

Minor:

1) The text in the figures are very very small. None of the y-axis of the bar graphs can be read.

Reply: We have slightly increased the font size and have uploaded higher resolution versions of the figures, which should permit reading of the y-axes. We can further increase the font size should the reviewer deem this helpful.

2) Maybe Fig 4 can be divided into two?

Reply: We have moved some of the data from Fig. 4 into the supplement and have split up the supplementary figures into more supplementary figures, hoping this improves the flow and presentation of the data.

3) Figure 5g can be enlarged, there is space in the figure.

Reply: We have re-arranged this figure and slightly enlarged panel 5g.

Thank you.

Reply: We thank the reviewer for their detailed comments and suggestions.

References

- 1 Monzon-Sandoval, J. *et al.* Lipopolysaccharide distinctively alters human microglia transcriptomes to resemble microglia from Alzheimer's disease mouse models. *Dis Model Mech* **15** (2022). <https://doi.org:10.1242/dmm.049349>
- 2 von Bartheld, C. S., Bahney, J. & Herculano-Houzel, S. The search for true numbers of neurons and glial cells in the human brain: A review of 150 years of cell counting. *J Comp Neurol* **524**, 3865-3895 (2016). <https://doi.org:10.1002/cne.24040>
- 3 Haenseler, W. *et al.* A Highly Efficient Human Pluripotent Stem Cell Microglia Model Displays a Neuronal-Co-culture-Specific Expression Profile and Inflammatory Response. *Stem Cell Reports* **8**, 1727-1742 (2017). <https://doi.org:10.1016/j.stemcr.2017.05.017>
- 4 Vahsen, B. F. *et al.* Human iPSC co-culture model to investigate the interaction between microglia and motor neurons. *Sci Rep* **12**, 12606 (2022). <https://doi.org:10.1038/s41598-022-16896-8>
- 5 Frakes, A. E. *et al.* Microglia induce motor neuron death via the classical NF-kappaB pathway in amyotrophic lateral sclerosis. *Neuron* **81**, 1009-1023 (2014). <https://doi.org:10.1016/j.neuron.2014.01.013>
- 6 Birger, A. *et al.* Human iPSC-derived astrocytes from ALS patients with mutated C9ORF72 show increased oxidative stress and neurotoxicity. *EBioMedicine* **50**, 274-289 (2019). <https://doi.org:10.1016/j.ebiom.2019.11.026>
- 7 Banerjee, P. *et al.* Cell-autonomous immune dysfunction driven by disrupted autophagy in C9orf72-ALS iPSC-derived microglia contributes to neurodegeneration. *Sci Adv* **9**, eabq0651 (2023). <https://doi.org:10.1126/sciadv.abq0651>
- 8 Lorenzini, I. *et al.* Moderate intrinsic phenotypic alterations in C9orf72 ALS/FTD iPSC-microglia despite the presence of C9orf72 pathological features. *Front Cell Neurosci* **17**, 1179796 (2023). <https://doi.org:10.3389/fncel.2023.1179796>
- 9 Levin, J. I. *et al.* The discovery of anthranilic acid-Based MMP inhibitors. Part 2: SAR of the 5-position and P11 groups. *Bioorganic & Medicinal Chemistry Letters* **11**, 2189-2192 (2001). [https://doi.org:https://doi.org/10.1016/S0960-894X\(01\)00419-X](https://doi.org:https://doi.org/10.1016/S0960-894X(01)00419-X)

REVIEWERS' COMMENTS

Reviewer #1 (Remarks to the Author):

The authors have addressed my questions and concerns in a thorough and appropriate fashion. I do not have further comments regarding the current version of the manuscript. I congratulate the authors for a very interesting study on the role of MMP9 in C9orf72-ALS pathophysiology and associated inflammation.

Reviewer #2 (Remarks to the Author):

The authors have added a substantial amount of additional work to a solid foundation.

The trans well assay, the use of recombinant MMP9 and the confirmation of RNA foci are especially nice additions.

Regarding the 1:1 ratio of microglia and neurons, the 1:1 ratio likely refers to all glia, not just microglia. Otherwise I think they have answered this reviewers points well.

I'd like to congratulate the authors on a fine piece of work. I recommend publication

Reviewer #3 (Remarks to the Author):

Understanding the basis of cell-autonomous and the non-cell autonomous factors that contribute to neuronal vulnerability and degeneration is of great importance. Especially within the context of C9orf72, one of the most common causes of neuronal degeneration in ALS, this topic still requires much investigation.

The authors took a very systemic approach and they also significantly improved their original submission both by performing additional experiments and by rewriting most critical sections of the manuscript.

The low n numbers had been a concern in the previous application and in some cases they are still limited to very small numbers and I think it was not possible for them to include more cases. This is reflected in the interpretation of their data and in their analyses. However, they have performed additional experiments to bring a better insight onto the mechanism and the mode of action and to the involvement of MMP9.

I like the manuscript in its current form and I think it adds tremendously to the field.

I have couple suggestions and would be happy if they were to do these before receiving "acceptance" for publication.

1) when discussing MMP9, I think it would really make sense to include C. Henderson's 2014 Neuron paper, which first revealed the potential role of MMP9 and its signaling in neuronal vulnerability and degeneration within the context of ALS. This would be a good fit when discussing in the discussion section.

2) Fig 1 a is exceptionally good for laying a foundation of the experimental design and makes Fig 1 very easy to follow. I think a similar schematic drawing can be included for Fig 2 to explain which cells are used which treatment is being compared. There is space in the figure.

3) In Fig 4a, the box says "patch clamping", but the experimental approach is high density electrode and it is not patch clamping. I think Fig 1 a info need to match experimental procedure

4) In fig 4h the immunocytochemistry for synaptophysin does not look like a synaptophysin immuno, which is usually puncta puncta. The image looks as if it is Tuj1 or NF-H immuno. Yes, puncta can also be seen if one looks carefully but overall image suggest a potential breach from a secondary or primary antibody. Can you please check this image and the immuno? Maybe you can use a better representative image.

5) Fig. 5 is a very significant figure and it tells two different stories and results. If there are no figure limitations, it would be better to divide it into two and to make a better emphasis on the mechanistic insight authors bring to the MMP9 mediated affects.

6) I think the discussion is well written, the text is much improved and I thank the authors for responding to all suggestions very carefully.

Thank you.

Response to the reviewers' comments

REVIEWER COMMENTS

Reviewer #1 (Remarks to the Author):

The authors have addressed my questions and concerns in a thorough and appropriate fashion. I do not have further comments regarding the current version of the manuscript. I congratulate the authors for a very interesting study on the role of MMP9 in C9orf72-ALS pathophysiology and associated inflammation.

Reply: We thank the reviewer for reviewing our manuscript and the positive feedback.

Reviewer #2 (Remarks to the Author):

The authors have added a substantial amount of additional work to a solid foundation.

The trans well assay, the use of recombinant MMP9 and the confirmation of RNA foci are especially nice additions.

Regarding the 1:1 ratio of microglia and neurons, the 1:1 ratio likely refers to all glia, not just microglia. Otherwise I think they have answered this reviewers points well.

I'd like to congratulate the authors on a fine piece of work. I recommend publication

Reply: We thank the reviewer for reviewing our manuscript and the positive feedback.

Reviewer #3 (Remarks to the Author):

Understanding the basis of cell-autonomous and the non-cell autonomous factors that contribute to neuronal vulnerability and degeneration is of great importance. Especially within the context of C9orf72, one of the most common causes of neuronal degeneration in ALS, this topic still requires much investigation.

The authors took a very systemic approach and they also significantly improved their original submission both by performing additional experiments and by rewriting most critical sections of the manuscript.

The low n numbers had been a concern in the previous application and in some cases they are still limited to very small numbers and I think it was not possible for them to include more cases. This is reflected in the interpretation of their data and in their analyses . However, they have performed additional experiments to bring a better insight onto the mechanism and the mode of action and to the involvement of MMP9.

I like the manuscript in its current form and I think it adds tremendously to the field.

I have couple suggestions and would be happy if they were to do these before receiving "acceptance" for publication.

Reply: We thank the reviewer for their time and the positive feedback and for providing further helpful suggestions. We provide a detailed response to the reviewer's comments below.

1) when discussing MMP9, I think it would really make sense to include C. Henderson's 2014 Neuron paper, which first revealed the potential role of MMP9 and its signaling in neuronal vulnerability and degeneration within the context of ALS. This would be a good fit when discussing in the discussion section.

Reply: We had already discussed the paper from Chris Henderson's lab in the discussion section (reference 22). We have now included a slightly expanded discussion of the paper (p. 19).

2) Fig 1 a is exceptionally good for laying a foundation of the experimental design and makes Fig 1 very easy to follow. I think a similar schematic drawing can be included for Fig 2 to explain which cells are used which treatment is being compared. There is space in the figure.

Reply: We agree with the reviewer and have added a small summary schematic to Fig. 2 (new Fig. 2a).

3) In Fig 4a, the box says "patch clamping", but the experimental approach is high density electrode and it is not patch clamping. I think Fig 1 a info need to match experimental procedure

Reply: We have indeed performed both patch clamping (the results are shown in both the main Figure 4d/e and the corresponding Supplementary Figure 7) and multi-electrode array recordings. We have therefore not removed 'patch clamping' from the summary schematic in Fig. 4a but instead added a box saying 'MEA'.

4) In fig 4h the immunocytochemistry for synaptophysin does not look like a synaptophysin immuno, which is usually puncta puncta. The image looks as if it is Tuj1 or NF-H immuno. Yes, puncta can also be seen if one looks carefully but overall image suggest a potential breach from a secondary or primary antibody. Can you please check this image and the immuno? Maybe you can use a better representative image.

Reply: This image is indeed from an ICC for both synaptophysin and TUJ1. We have replaced the images in both Fig. 4h and Supp. Fig. 9b with composite images showing both the synaptophysin and TUJ1 staining (similar to Fig. 2a in our previous publication¹), which better demonstrates the specific punctate expression pattern of the synaptophysin staining.

5) Fig. 5 is a very significant figure and it tells two different stories and results. If there are no figure limitations, it would be better to divide it into two and to make a better emphasis on the mechanistic insight authors bring to the MMP9 mediated affects.

Reply: We agree with the reviewer and have split figure 5 into two figures – the new figure 6 shows the results after pro-longed stimulation with LPS.

6) I think the discussion is well written, the text is much improved and I thank the authors for responding to all suggestions very carefully.

Reply: We thank the reviewer for their positive feedback.

Thank you.

Reply: We thank the reviewer for their time dedicated to the review of our manuscript.

References

- 1 Vahsen, B. F. *et al.* Human iPSC co-culture model to investigate the interaction between microglia and motor neurons. *Sci Rep* **12**, 12606 (2022). <https://doi.org:10.1038/s41598-022-16896-8>